# Social Human Robot Embodied Conversation (SHREC) Dataset: Benchmarking Foundational Models' Social Reasoning

## Abstract

Our work focuses on the social reasoning capabilities of foundational models for real-world human–robot interactions. We introduce the Social Human Robot Embodied Conversation (SHREC) Dataset, a benchmark of ∼400 real-world human-robot interaction videos and over 10K annotations, capturing robot social errors, competencies, underlying rationales, and corrections. Unlike prior datasets focused on human–human interactions, the SHREC Dataset uniquely highlights the social challenges faced by real-world social robots such as emotion understanding, intention tracking, and conversational mechanics. Moreover, current foundational models struggle to recognize these deficits, which manifest as subtle, socially situated failures. To evaluate AI models' capacity for social reasoning, we define eight benchmark tasks targeting critical areas such as (1) detection of social errors and competencies, (2) identification of underlying social attributes, (3) comprehension of interaction flow, and (4) providing rationale and alternative correct actions. Experiments with state-of-the-art foundational models, alongside human evaluations, reveal substantial performance gaps—underscoring the difficulty and providing directions in developing socially intelligent AI.

## 1 Introduction

In this work, we aim to advance the social reasoning capabilities of physically embodied AI agents, particularly tabletop social robots, engaged in real-world, socially interactive conversations. To this end, we focus on understanding and modeling both social competencies (i.e. desirable behaviors) and social errors (i.e. norm violations or failures in interaction) that arise during natural human-robot interactions. While prior work in social intelligence has primarily centered on human–human interaction datasets (Mathur et al., 2025; Zadeh et al., 2019a; Wilf et al., 2023), these settings do not capture the unique challenges that arise when robots interact with humans. Unlike humans, embodied AI agents may have varying and lacking socio-cognitive skills such as emotion understanding, belief tracking, or conversational coordination (Lake et al., 2017; Deng et al., 2023; Bhattacharyya & Wang, 2025; Arora et al., 2025), leading to failure cases that are subtle, socially situated, and poorly represented in existing resources.

To this end, we introduce the **Social Human Robot Embodied Conversation (SHREC) Dataset**, a dataset of 400+ videos capturing natural conversational interactions between a human and a physical tabletop social robot, making it, to the best of our

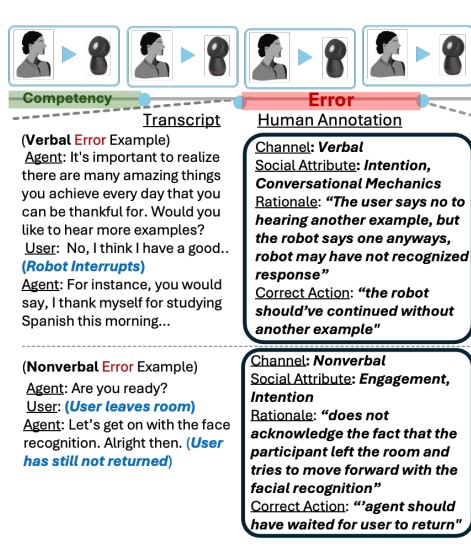

Figure 1: SHREC Dataset dataset offers real-world **Social Human Robot Embodied Conversation** videos and annotations of errors and competencies, the channel and type of social attribute, along with rationale and possible corrective actions. (Top) Error sourced from verbal (audio) channel, (Bottom) Error sourced from non-verbal (visual) channel.

knowledge, one of the largest real-world human–social robot interaction benchmark dataset available to date (see App. Tab. 4). We focus on a tabletop social robot, which is a widely used class of physically embodied agents, as a principled starting point for studying real-world social reasoning, and introduce a benchmark dataset that addresses the scarcity of real-world social robot interaction data available for advancing research in this area. The dataset is accompanied by 10K+ human annotations of the robot's social errors (undesirable behaviors) competencies (desirable behaviors), rationale and corrective actions for the AI agent. These annotations are grounded from prior HRI taxonomies (Tian & Oviatt, 2021; Fitrianie et al., 2022) of seven key social attributes that underpin social interactions, emotion, engagement, conversational mechanics, knowledge state, intention, social relationships, and norms.

Our dataset offers a novel resource to assess foundational models' capabilties in identifying a social robot's social errors and competencies in real-world human-robot interactions, filling a gap not addressed by current social intelligence benchmarks. To systematically evaluate social reasoning of state-of-the-art AI models, we propose eight benchmark tasks, spanning four core dimensions: (1) errors and competency detection in the robot's behavior, (2) social attribute identification related to the errors and competencies, (3) interaction progression reasoning, and (4) rationale and correction reasoning (outlined in Sec. 3). Beyond assessing models' social reasoning, generalization, and robustness in real-world, multimodal, and socially grounded settings (Zhou et al., 2024; Davis & Marcus, 2015; Lake et al., 2017; Sap et al., 2019a; Ross et al., 2022; Ludan et al., 2023), these tasks also serve as structured probes for fundamental representation learning challenges. They require multimodal alignment to integrate visual, auditory, and linguistic signals into representations for social understanding; reasoning to infer how specific behaviors influence downstream interaction trajectories; and reward-learning to align embodied agents to human social preferences. Our benchmark provides not only a principled testbed for advancing socially intelligent foundation models but also a new resource investigate these core learning problems.

We benchmarked 17 state-of-the-art LLMs and VLMs, including ChatGPT (Hurst et al., 2024) and Gemini (Team et al., 2024) variants, on these tasks. While some models excel in specific subtasks, none perform uniformly well across the board. Notably, the gap between model and human performance remains substantial, underscoring the challenge and novelty of our benchmark. These findings highlight Social Human Robot Embodied Conversation (SHREC) Dataset as a valuable testbed for diagnosing and improving the social reasoning abilities of embodied AI agents, and for guiding the development of reward models and evaluators aligned with social intelligence (Ouyang et al., 2022; Zhou et al., 2024; Lee et al., 2023; Chen et al., 2024b; Zheng et al., 2023).

## 2 SHREC: DATASET OF HUMAN ROBOT SOCIAL EMBODIED CONVERSATION

Our dataset consists of 10,353 annotations from 403 interaction videos spanning over 3,500 minutes. Under a newly accepted IRB protocol, we annotated and anonymized data on three prior human-robot interaction studies (Shen et al., 2024; Jeong et al., 2023b; 2020) to be shared for dissemination. To enable public access while preserving participant privacy, we follow institutional IRB procedures and release the dataset under gated access. All personally identifiable information (PII) in transcripts is filtered. For video anonymization, we leverage FRESCO (Yang et al., 2024), a zero-shot video-to-video diffusion framework, to perform stylized face transfer. FRESCO's spatial-temporal consistency allows us to reliably replace participant faces while maintaining coherence across frames. This ensures that the social signals (e.g., gaze, affect) critical for interaction analysis are preserved while protecting individual identities through high-fidelity anonymization. n selecting FRESCO, we compared several anonymization techniques on the same interaction clips, generating matched variants from the raw videos and evaluating them with the identical pipeline (see Appendix I). Most vision-language models (VLMs) operate at 1 Hz, which is the default temporal resolution used in our benchmark. Hence, we release video frames sampled at 1 Hz to align with common VLM processing rates and ensure compatibility across models. [1]

We describe the three real-world studies that form the foundation of the SHREC dataset: **Empathic++** (Shen et al., 2024): a ChatGPT-powered social robotic agent acts as an empathic companion, facilitating the exchange of emotionally meaningful stories using narrative therapy techniques. The goal

---

[1]To support community interest in fine-grained temporal analysis, we will release an extended addendum with 15Hz videos (processing is underway and sample videos are included in the supplementary). We maintain a private hold-out set of ∼40 videos to prevent data contamination and enable future benchmarking and evaluations.

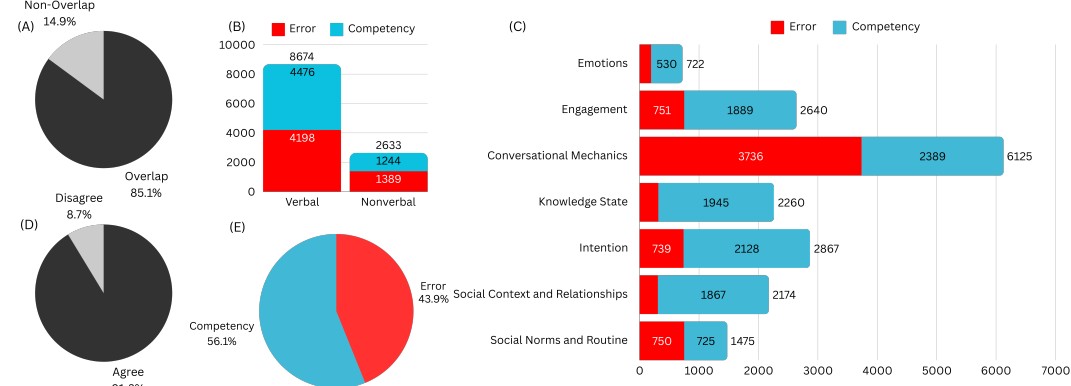

Figure 2: SHREC Dataset contains **high overlapping** annotations with a **high level of agreement**. The dataset includes error and competency labels, and annotations for the source of evidence either from nonverbal cues, verbal cues, and explanatory factors in the form of seven key social attributes.

is to enhance users' feelings of connection and belonging through emotionally attuned interaction. **Wellness-Dorm** (Jeong et al., 2020): A socially assistive robot was deployed as a positive psychology coach for college students living in on-campus dormitories. The robot was manually scripted with seven intervention types grounded in established principles of positive psychology, such as gratitude, strengths-based reflection, and goal setting. **Wellness-Home** (Jeong et al., 2023b): Positive psychology robots were deployed in participants' homes. All three datasets are ready for dissemination (We kindly recommend the reviewer to view the supplementary materials for examples of the dataset).

## 2.1 HUMAN ANNOTATIONS

As shown in Fig. 1 and App. H, annotators watch videos of conversational human-robot interactions and are asked to detect segments which manifest a ***social error or competency***. ***Social Competence*** is defined as the behaviors where the agent successfully conduct social interactions by being aware of and identifying social-emotional cues, processing such cues, and expressing a user-expected response to these cues (Halberstadt et al., 2001). ***Social Error*** is defined as the behaviors where the robot deviates from the desired behaviors expected by a user and degrades the user's perception of a robot's social competence (Tian & Oviatt, 2021). Then, they are asked to identify which ***social attribute*** the segment is related to (described in Sec. 2.1.1) and offer a ***rationale*** of why they believe so. If the segment was an error, they are asked to suggest an alternative ***correct action***.

### 2.1.1 SOCIAL ATTRIBUTES

Given a segment of social error or competencies, we are interested in the *explanatory factors* related to a social error or competency. To do so, we consider seven specific categories of social attributes that are related to social errors and competencies. The definitions for each of the attributes are as follows: **Emotions:** The ability to identify and interpret emotional expressions in oneself and others, allowing for empathetic responses and social awareness, e.g., recognizing that someone crying might mean they're sad (Golan et al., 2006). **Engagement:** The skill to observe and assess levels of participation and involvement in social interactions, including cues that indicate interest or disinterest, e.g., continuing to tell a story when a listener is engaged (Davis, 1980). **Conversational Mechanics:** Understanding the structure and flow of conversations, including turn-taking, interruptions, and cues for when to speak or listen, e.g., waiting for another person to finish speaking before taking a turn (Fusaroli & Tylén, 2016). **Knowledge State:** The ability to assess what others know or believe, as well as being aware of one's own knowledge in social situations, e.g., make reference to a user's dog recalling that the user has a dog (Baron-Cohen et al., 1999). **User Intention:** The capacity to infer the goals or purposes behind the actions and words of others, facilitating better responses in social interactions, e.g., when the user says "I'll be right back", indicates that the user will vacate and then return (Dziobek et al., 2006). **Social Context and Relationships:** The ability to identify and understand the dynamics of social relationships and the context in which they occur, influencing behavior and expectations, e.g., knowing how to act in front of a close friend vs. colleague at work (Baron-Cohen et al., 1999). **Social Norms and Routines:** The skill to identify accepted behaviors and attitudes within a social group, as well as recognizing negative or harmful interactions that violate these norms. e.g., understanding that waving hands at the beginning of the interaction is a sign of a greeting (Thoits, 2004)

Figure 3: Our benchmark offers eight tasks dedicated to probing four core facets of AI model's social reasoning: (1) detecting social errors and competencies, (2) identifying social attributes, (3) understanding the flow of social interactions, and (4) rationalization and correction of social errors.

## 2.2 DATASET STATISTICS

We refer the reader to Figure 2 for the overall statistics of our Social Human Robot Embodied Conversation (SHREC) Dataset. As shown in Figure 2-(A), 85.1% of the dataset consists of overlapping annotations, where multiple annotators independently labeled the same segment of an interaction. Amongst the overlapping samples as shown in Figure 2-(D), we find a 91.3% overall agreement, where annotators agree on the error/competency labels. In Figure 2-(B), we find that more annotations come from verbal channel, rather than the non-verbal channel. As shown in Figure 2-(E), we find 56% corresponds to competency labels and 43% corresponds to error labels. In Figure 2-(C), we display the social attributes the annotators have selected, and we find that the most number of annotations belong to the conversational mechanics category, followed by intention and engagement. **Annotator Consistency:** To ensure consistent annotations, we recruited and trained 3 annotators. They were provided a set of definitions and annotation guidelines as shown in Appendix H. The annotators met multiple times to discuss edge scenarios and ambiguous segments. After annotating independently, annotators cross-validated each other's work for agreement. Some scenarios that were too subjective to reach full consensus yielded some annotations in the Disagree category of 8.7%.

## 3 TASKS & EXPERIMENTS

We developed eight tasks to measure the social reasoning capability of foundational models. Each task measures a different social reasoning capability of the model, spanning from identifying social errors and competencies, to reasoning about the errors and offering correct actions. Below, we describe four main research questions which motivated the design of the social reasoning tasks and the corresponding tasks which serve to address these questions.

**RQ1: Can AI models be used as an automatic evaluator of social interactions?**
(1) Errors and Competence Detection, (2) Error Detection (Sec. 3.1, 4.1)

**RQ2: Can AI models identify the explanatory factors associated with social errors and social competencies?**
(3) Social Attribute Identification, (4) Multiple Social Attribute Presence (Sec. 3.2, 4.2)

**RQ3: Do AI models understand the sequential contingencies or the "flow" of social interactions?**
If-Then Reasoning: (5) Pre-Condition and (6) Post-Condition (Sec. 3.3, 4.3)

**RQ4: Can AI models recognize the reasons of errors and infer the correct action?**
(7) Rationale and (8) Correction Reasoning (Sec. 3.4, 4.4)

We utilize LLMs & VLMs and formulate the tasks in the following manner. We treat a foundational model as $\pi$. We define relevant contextual information (images and transcript) and the task-specific question as query $Q$. Formally, we define $Q = \{q_1, q_2, \ldots, q_n\}$ as the set of tokens representing the question and the transcript (including video or image frames if multimodal LLM). For the image-based models, we provide 15 uniformly sampled frames from the video segment. For video-based models, we feed in the raw video as input and rely on model-specific preprocessing steps. The output of the model is $\pi(Q) = O$, where $O$ is the set of tokens representing the output of the LLM. We denote the ground-truth answer for each task as $A$, which can take the form of a multiple-choice option $(A, B, C, D, E)$, a Boolean value (True/False). Since $O$ is often a free-form string, we apply a post-processing step using Pydantic (Colvin et al., 2025), where another LLM coerces $O$ into

the discrete answer space of the task (e.g., mapping "The correct choice is option C" → "C"). We then define correctness as Correctness$(A, O) = 1$ if Pydantic$(O) = A$, $0$ otherwise. This definition ensures that correctness measures exact agreement with the ground truth rather than string overlap. **Human Comparison:** For fairness, we asked human annotators to perform the same tasks as the LLMs/VLMs (given identical prompts as instructions). Unlike the original annotations, where annotators watched videos and marked salient error/competency regions, here the annotator viewed the pre-segmented data and completed the same task as the models. We then compared their responses to the original ground-truth labels, yielding a consistent estimate of human-level performance under identical constraints.

## 3.1 ERROR AND COMPETENCE

We use *Error Detection and Competence Detection* as a proxy to evaluate whether AI models can effectively serve as automatic evaluators of social interactions (Ouyang et al., 2022; Zhou et al., 2024; Lee et al., 2023; Chen et al., 2024b; Zheng et al., 2023). The model is provided with a prompt which includes an interaction segment between a user and a social robot. The model then predicts whether the sequence contains a social error, competence, or neither. For the *Error Detection* task, the model is only required to determine whether a given instance constitutes an error or not. For these tasks, we evaluate with accuracy and macro-F1 (unweighted average of F1) scores. These prompts are shown below:

> **Social Error, Competence, None Detection (Error/Comp./None):** We provide the interaction between social agent and a user: {*Interaction Transcript*} Does the agent exhibit (A) Social Competence or (B) Social Error or (C) None?
>
> **Social Error Detection (Error):** We provide the interaction between social agent and a user: {*Interaction Transcript*} Does the agent exhibit (A) Social Error or (B) No Social Error?

## 3.2 SOCIAL ATTRIBUTE

Additionally, if a sequence is identified as either a social error or competence, we further determine the explanatory factor, i.e. which specific *social attribute* it was associated with. This helps assess whether AI models can provide more detailed evaluations by identifying specific types of social attributes. Furthermore, fine-grained feedback in the form of attribute-specific reward signals allows us to disentangle different dimensions of social behavior—such as emotional response, conversational mechanics, or knowledge state, thereby enabling more interpretable and targeted model improvements (Wu et al., 2023b). By aligning rewards with these distinct social attributes, we can not only evaluate whether a model exhibits socially competent behavior but also pinpoint which aspect it succeeded or failed in, facilitating modular training and fine-tuning strategies. Using the same context as Section 3.1, LLMs and VLMs are further provided with the label indicating whether the instance is a social error or competence, and the model predicts the relevant attribute(s) as defined in Section 2. This task can be viewed as a multi-label classification problem, as multiple social attributes may co-occur in a single instance. For example, a sequence might be annotated with both conversational mechanics (e.g., a delayed response) and knowledge state (e.g., the robot forgetting the user's name). To evaluate whether models can detect instances associated with more than one social attribute, we introduce the multiple social attribute detection task. Below, we show the prompt for this task:

> **Social Attribute Identification (Attr.):** We provide the interaction between social agent and a user: {*Interaction Transcript*}. This segment corresponds to an {*Error or Competence*} in social behavior. Which of the following categories is this segment related to? (A) Emotions, (B) Engagement, (C) Conversational Mechanics, (D) Knowledge State of Others and Self, (E) Understanding Intention of the User, (F) Social Context and Relationships, (G) Social Norms and Routines.
>
> **Multiple Social Attribute Presence (Multi. Attr.):** We provide a transcript of an interaction between the social agent and a user: {*Interaction Transcript*}. The agent's behavior in this interaction corresponds to {*Error or Competence*}. Consider the following seven social attributes: [Same as Above]. Based on the transcript, determine whether the agent's behavior involves multiple social attributes. Respond with "True" if the behavior demonstrates more than one social attribute. Respond with "False" otherwise.

For the task of social attribute identification, we evaluate with accuracy and macro-F1 (F1) scores. As there can be more than a single attribute label associated with a sample, we further report Partial Match (PM) to evaluate the proportion of instances where the model correctly predicts at least one of the true labels. For multiple social attribute detection, we evaluate with accuracy and macro-F1.

### 3.3 IF-THEN REASONING

We test whether or not AI models can understand sequential contingencies or the flow of social interactions, by testing if they can predict probable pre-and-post conditions of a given competent social interaction, otherwise known as *if-then reasoning*. Such inferential reasoning tasks, determine whether models have learnt spurious task-specific correlations or more generalizable reasoning (Davis & Marcus, 2015; Lake et al., 2017; Sap et al., 2019a). We formulate two tasks which checks for the pre-conditons and post-conditions.

> **Interaction Flow (Pre-Condition\Post-Condition):** We provide the agent\user's behavior: {*Agent\User Transcript*}. From the following choices of the user\agent's behaviors: (1) {*User\Agent Transcript 1*}, (2) {*User\Agent Transcript 2*}, (3) {*User\Agent Transcript 3*}, (4) {*User\Agent Transcript 4*}, (5) {*User\Agent Transcript 5*}, select which user\agent's behavior was the appropriate response before\after the agent\user's action.

For *pre-condition*, given the agent's response, the model must identify the plausible pre-condition, i.e., the user's behavior prior to the agent's utterance. Vice versa, for *post-condition*, given the user's response, the model must predict the plausible post-conditions, i.e., the agent's actions after the event. These tasks are set-up as a multiple choice Q&A set up, where they are asked to predict the correct choice of post or pre-condition transcript. To acquire incorrect answer choices, first, we remove any samples that share the same transcript as the correct answer, if such samples exist. Next, we randomly select four other samples and extract the relevant transcript (either the user's or the agent's utterance) which serve as incorrect options. Finally, we shuffle the correct answer among the options to ensure its position is randomized. For these tasks, we use accuracy scores for evaluation.

### 3.4 RATIONALE AND CORRECTION

We formulate two tasks, *rationale*, which require models to provide rationale on why the segment contains a social error and *correction*, requiring models to suggest what would have been the correct action. These tasks probes the model's ability to not just diagnose an interaction segment but also to explain why the detected behavior is incorrect or inappropriate and identify a correct alternative action. Hence, they are directly tied to evaluating the social reasoning capabilities of AI models. Furthermore, it is well-known that models and robots that can provide rationales and correct actions enhance their robustness (Ross et al., 2022; Ludan et al., 2023) and trustworthiness (Kox et al., 2021; Javaid & Estivill-Castro, 2021; Esterwood & Robert, 2025).

> **Rationale:** We provide the interaction between social agent and a user, which corresponds to an error in social behavior: {*Interaction Transcript*}. Select which is the correct rationale behind the error. (1) {*rationale 1*} (2) {*rationale 2*} (3) {*rationale 3*} (4) {*rationale 4*} (5) {*rationale 5*}
>
> **Correction:** We provide the interaction between social agent and a user, which corresponds to an error in social behavior: {*Interaction Transcript*}. From the following choices select which behavior the social agent should have done instead. (1) {*Correction 1*} (2) {*Correction 2*} (3) {*Correction 3*} (4) {*Correction 4*} (5) {*Correction 5*}

For the *rationale* and *correction* task, models are given interaction segments corresponding to social error. Then, for the *rationale* task, the model is asked to predict the correct reason for the error. For the *correction* task, the model must choose the alternate correct action. Both tasks are set-up as multiple choice Q&As, we provide five answer choices: one ground-truth option and four incorrect choices sampled from other instances. To ensure the incorrect choices are distinct, we only select samples with different social attribute annotations than that of the answer sample. These tasks are evaluated with accuracy.

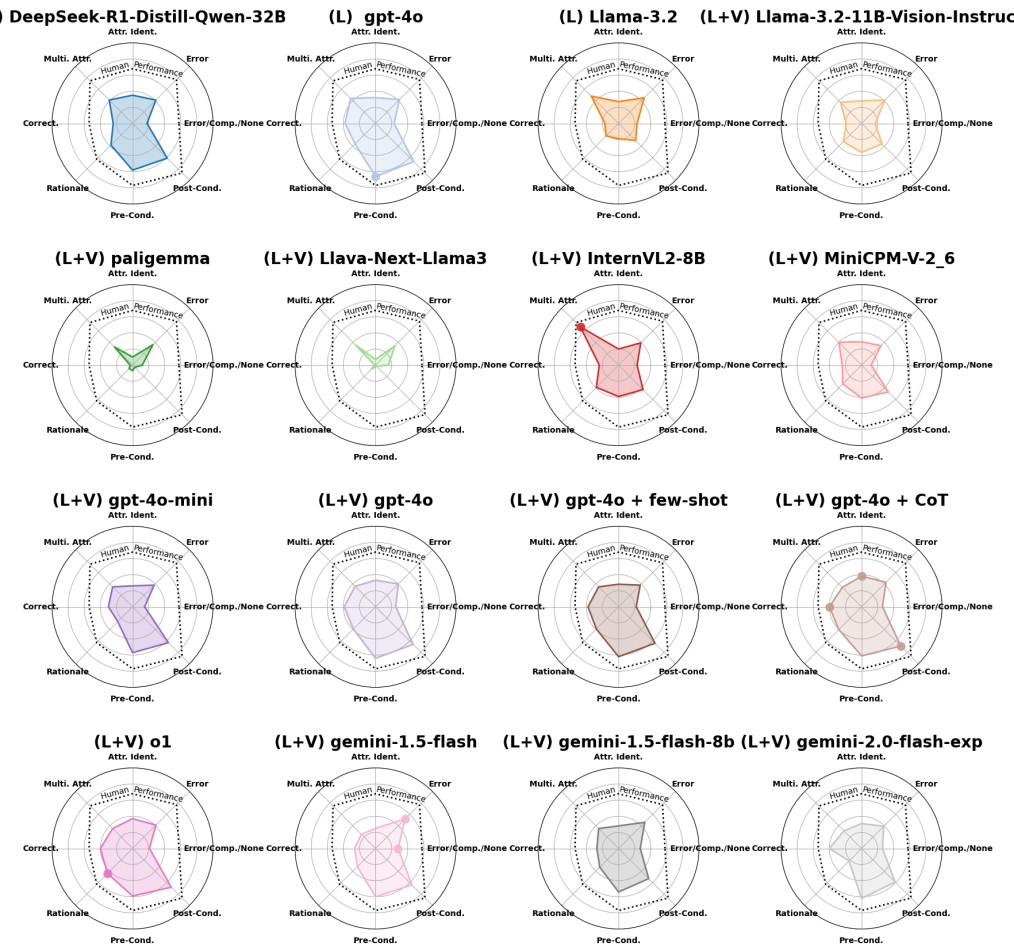

Figure 4: Results per model across all 8 tasks. Human performance is marked in dashed lines. (L): language-only inputs, (L+V): language and visual inputs. ***Models from the same family tend to show similar shapes on the radar plot, reflecting consistent reasoning patterns across capabilities, which supports the benchmark's sensitivity to measure underlying social reasoning abilities.***

## 4 RESULTS & DISCUSSION

We evaluate the performance of 17 language and vision-language models (VLMs), including DeepSeek-R1-Qwen-32B (Guo et al., 2025), gpt-4o-mini, gpt-4o (Hurst et al., 2024), Llama-3.2, Llama-3.2-Vision-Instruct (Grattafiori et al., 2024), paligemma (Beyer et al., 2024), Llava-Next-Llama3 (Zhang et al., 2024), InternVL2-8B (Chen et al., 2024a), MiniCPM (Yao et al., 2024), o1 (Jaech et al., 2024), gemini-1.5 (Team et al., 2024) and gemini-2.0 (Pichai et al., 2024) and their variants on our benchmark tasks. Fig. 4 summarizes the evaluation results. No single model excels across all social reasoning tasks, underscoring the need for advancements in training FMs for social reasoning.

### 4.1 RQ1: CAN AI MODELS BE USED AS AUTOMATIC EVALUATORS OF SOCIAL INTERACTIONS? [TAB. 1-LEFT]

We refer readers to the left side of Table 1, which presents results for the *Error/Competence/None Detection* task. The best-performing models are gemini-1.5-flash using both language and video inputs, achieving 0.32 accuracy and 0.28 F1. For the binary *Error Detection* task, the same model achieve higher performance—0.55 exact match and 0.52 F1—suggesting that models are more effective at detecting errors alone than distinguishing among all three categories. Nonetheless, the overall performance remains modest, reflecting the task's complexity and the models' difficulty in capturing the nuances of social error and competence. Appendix F further analyzes model

| Model Name | Errors and Competence | | | | Social Attribute | | | | |
|---|---|---|---|---|---|---|---|---|---|
| | Error/Comp./None | | Error | | Attr. Identification | | | Multi. Attr. Presence | |
| | Acc. | F1 | Acc. | F1 | Acc. | F1 | Partial | Acc. | F1 |
| (L) DeepSeek-R1-Distill-Qwen-32B | 0.22±0.02 | 0.18±0.02 | 0.51±0.02 | 0.41±0.01 | 0.02±0.01 | **0.35±0.01** | 0.67±0.02 | 0.52±0.01 | 0.41±0.03 |
| (L) gpt-4o-mini | 0.26±0.02 | 0.21±0.02 | 0.52±0.02 | 0.46±0.01 | 0.01±0.01 | 0.17±0.01 | 0.41±0.04 | 0.46±0.02 | 0.40±0.03 |
| (L) gpt-4o | 0.25±0.02 | 0.23±0.02 | 0.50±0.02 | 0.42±0.02 | 0.02±0.01 | 0.20±0.01 | 0.49±0.03 | 0.52±0.04 | 0.44±0.06 |
| (L) Llama-3.2 | 0.26±0.01 | 0.23±0.00 | 0.50±0.02 | 0.45±0.02 | 0.02±0.02 | 0.18±0.04 | 0.43±0.07 | 0.55±0.04 | 0.48±0.04 |
| (L) Llama-3.2-11B-Vision-Instruct | 0.22±0.04 | 0.17±0.03 | 0.50±0.05 | 0.41±0.04 | 0.01±0.01 | 0.24±0.01 | 0.56±0.03 | 0.50±0.02 | 0.37±0.03 |
| (L+V) paligemma | 0.19±0.02 | 0.12±0.01 | 0.49±0.02 | 0.36±0.02 | 0.00±0.00 | 0.11±0.02 | 0.27±0.04 | 0.53±0.03 | 0.32±0.02 |
| (L+V) Llava-Next-Llama3 | 0.19±0.03 | 0.16±0.03 | 0.49±0.03 | 0.34±0.02 | 0.01±0.01 | 0.09±0.02 | 0.25±0.03 | 0.51±0.03 | 0.35±0.02 |
| (L+V) InternVL2-8B | 0.24±0.04 | 0.23±0.04 | 0.49±0.04 | 0.39±0.04 | 0.01±0.01 | 0.21±0.02 | 0.51±0.06 | **0.67±0.02** | **0.67±0.02** |
| (L+V) MiniCPM-V-2_6 | 0.18±0.02 | 0.12±0.01 | 0.48±0.03 | 0.34±0.02 | 0.00±0.00 | 0.27±0.02 | 0.58±0.04 | 0.51±0.01 | 0.40±0.08 |
| (L+V) gpt-4o-mini | 0.20±0.03 | 0.15±0.03 | 0.49±0.02 | 0.38±0.01 | 0.03±0.03 | 0.17±0.01 | 0.42±0.02 | 0.50±0.01 | 0.35±0.01 |
| (L+V) gpt-4o | 0.26±0.03 | 0.25±0.03 | 0.50±0.02 | 0.40±0.02 | 0.01±0.01 | 0.21±0.02 | 0.51±0.04 | 0.49±0.01 | 0.36±0.01 |
| (L+V) gpt-4o + few-shot | 0.24±0.02 | 0.22±0.01 | 0.48±0.02 | 0.38±0.01 | 0.01±0.01 | 0.21±0.02 | 0.52±0.04 | 0.49±0.01 | 0.35±0.02 |
| (L+V) gpt-4o + cot | 0.27±0.03 | 0.26±0.03 | 0.50±0.02 | 0.43±0.02 | 0.04±0.01 | 0.33±0.02 | 0.64±0.06 | 0.49±0.01 | 0.34±0.01 |
| (L+V) o1 | 0.24±0.04 | 0.21±0.04 | 0.50±0.02 | 0.41±0.02 | 0.02±0.02 | 0.32±0.02 | 0.67±0.02 | 0.49±0.01 | 0.35±0.02 |
| (L+V) gemini-1.5-flash | **0.32±0.01** | **0.28±0.02** | **0.55±0.02** | **0.52±0.01** | 0.01±0.02 | 0.17±0.02 | 0.43±0.05 | 0.45±0.03 | 0.25±0.02 |
| (L+V) gemini-1.5-flash-8b | 0.28±0.02 | 0.27±0.02 | 0.51±0.02 | 0.46±0.02 | 0.03±0.02 | 0.24±0.02 | **0.72±0.04** | 0.52±0.02 | 0.35±0.02 |
| (L+V) gemini-2.0-flash-exp | 0.27±0.05 | 0.26±0.04 | 0.49±0.03 | 0.39±0.02 | 0.01±0.00 | 0.28±0.01 | 0.62±0.03 | 0.47±0.03 | 0.31±0.02 |
| Human | 0.57 | 0.45 | 0.77 | 0.71 | 0.24 | 0.67 | 0.76 | 0.57 | 0.75 |

Table 1: Results for **Errors and Competence** detection tasks and **Social Attribute** identification tasks, with ± indicating one standard deviation. (L): language-only, (L+V): language and vision.

performance by social attribute, revealing that some models excel in detecting specific types of errors. These findings underscore the gap between current AI capabilities and human-level social reasoning, pointing to the need for continued research in this area.

### 4.2 RQ2: CAN AI MODELS IDENTIFY THE EXPLANATORY FACTORS ASSOCIATED WITH SOCIAL ERRORS AND SOCIAL COMPETENCIES? [TAB. 1-RIGHT]

In Table 1, under the column *Attr. Identification*, we evaluate the ability of foundational models to identify explanatory factors in the form of social attributes. We report accuracy, F1, and partial accuracy (i.e., predicting at least one attribute correctly). Notably, most models perform well on partial accuracy, demonstrating the ability to identify at least one relevant attribute. In particular, Deepseek-R1-Distill-Qwen-32B, gemini-1.5-flash-8b, gemini-2.0, o1, and several gpt-4o variants with image input and chain-of-thought (CoT) reasoning achieve partial accuracy scores above 0.60, highlighting their relative strength in capturing aspects of social attribute prediction. However, all models struggle to predict the full and correct set of attributes, as reflected in low accuracy and modest F1 scores, which indicates FMs remain limited in handling the complex, multi-label, and co-occurring nature of social attribute classification. Human evaluation further reveals that, despite a clear performance gap, this remains a challenging task even for humans. To further probe these limitations, we refer readers to the right-most columns of Table 1, where we assess whether models can detect the presence of multiple attributes within a segment. Interestingly, InternVL2 achieves the highest accuracy and F1 in this setting. However, the majority of models perform at or below 0.5, indicating persistent difficulty in recognizing multi-attribute cases—a key failure mode in this task. Additional analysis is provided in Appendix C, which examines the role of subjectivity and co-occurrence, as social attributes often co-occur in a single segment and are subject to perceiver-dependent interpretations of social constructs (Searle, 1998; Mathur et al., 2024). Furthermore, in Appendix G, Fig. 9, we carry out further analysis to identify which attributes are easier for error detection.

### 4.3 RQ3: DO AI MODELS UNDERSTAND THE SEQUENTIAL CONTINGENCIES OR THE "FLOW" OF SOCIAL INTERACTIONS? [TAB. 2-LEFT]

Many large language models, including early versions of BERT (Devlin et al., 2019) and ALBERT (Lan et al., 2019), were trained with next sentence prediction (NSP) and sentence ordering objectives, which aim to model discourse coherence and temporal continuity by predicting whether one sentence logically follows another. This objective aligns with the structure of *if-then reasoning* tasks, partially explaining the relatively strong performance of language-only models—particularly the gpt-4o variants, which achieve 0.66 on *pre-condition* and 0.69 on *post-condition* inference. However, their performance still lags behind humans. Despite progress in temporal reasoning (Sap et al., 2019a), we observe a significant gap between nascent open-source vision-language models (VLMs) and even text-only LMs, suggesting that visual input alone does not guarantee better reasoning about interactional contingencies.

### 4.4 RQ4: CAN AI MODELS RECOGNIZE THE REASONS OF ERRORS AND INFER THE CORRECT ACTION? [TAB. 2-RIGHT]

The *rationale* and *correction* tasks evaluate a model's ability to interpret interaction context, infer the cause of social errors or competencies, and suggest appropriate alternative correct actions, making them key indicators of social reasoning. These are among the most challenging tasks in the dataset, as reflected by human-level difficulty. For the **rationale** task, the `o1` model performs best with a score of 0.44, followed by `gpt-4o` with Chain-of-Thought (CoT) at 0.40. For the **correction** task, `o1`, `gpt-4o` with CoT, and `gemini-2.0-flash` again perform well, each achieving a score of 0.40. These trends are consistent with recent advancements in reasoning, where best-performing models like `o1`, `gpt-4o` with CoT, and `gemini-2.0-flash` are specifically trained to handle complex reasoning tasks (Team et al., 2023; OpenAI, 2023).

|  | If-Then Reasoning | | Rationale & Correct. | |
|---|---|---|---|---|
|  | Pre-Condition | Post-Condition | Rationale | Correction |
| **Model Name** | Acc. | Acc. | Acc. | Acc. |
| (L) DeepSeek-R1-Distill-Qwen-32B | 0.58±0.03 | 0.61±0.02 | 0.38±0.05 | 0.24±0.01 |
| (L) gpt-4o-mini | 0.58±0.02 | 0.62±0.03 | 0.29±0.04 | 0.31±0.08 |
| (L) gpt-4o | **0.66±0.00** | 0.67±0.02 | 0.36±0.01 | 0.38±0.04 |
| (L) Llama-3.2 | 0.19±0.08 | 0.30±0.01 | 0.22±0.02 | 0.17±0.02 |
| (L) Llama-3.2-11B-Vision-Instruct | 0.36±0.05 | 0.36±0.05 | 0.32±0.05 | 0.19±0.02 |
| (L+V) paligemma | 0.07±0.02 | 0.04±0.03 | 0.06±0.05 | 0.03±0.01 |
| (L+V) Llava-Next-Llama3 | 0.02±0.01 | 0.02±0.01 | 0.07±0.02 | 0.01±0.01 |
| (L+V) InternVL2-8B | 0.39±0.09 | 0.43±0.02 | 0.39±0.05 | 0.24±0.03 |
| (L+V) MiniCPM-V-2_6 | 0.41±0.07 | 0.47±0.02 | 0.33±0.05 | 0.24±0.05 |
| (L+V) gpt-4o-mini | 0.57±0.05 | 0.63±0.04 | 0.26±0.04 | 0.30±0.06 |
| (L+V) gpt-4o | 0.64±0.05 | 0.66±0.05 | 0.39±0.08 | 0.39±0.08 |
| (L+V) gpt-4o + few-shot | 0.62±0.04 | 0.64±0.03 | 0.39±0.04 | 0.38±0.04 |
| (L+V) gpt-4o + cot | 0.61±0.05 | **0.69±0.05** | 0.40±0.03 | **0.40±0.05** |
| (L+V) o1 | 0.59±0.08 | 0.68±0.02 | **0.44±0.10** | 0.40±0.03 |
| (L+V) gemini-1.5-flash | 0.60±0.10 | 0.64±0.07 | 0.32±0.04 | 0.26±0.07 |
| (L+V) gemini-1.5-flash-8b | 0.54±0.09 | 0.53±0.06 | 0.33±0.06 | 0.27±0.04 |
| (L+V) gemini-2.0-flash-exp | 0.62±0.07 | 0.59±0.09 | 0.22±0.04 | **0.40±0.05** |
| Human | 0.77 | 0.87 | 0.63 | 0.54 |

Table 2: **If-Then Reasoning** & **Rationale & Correction** Results; ± indicates one std. dev. (L): language-only, (L+V): language & vision.

## 5 RELATED WORK

**Datasets for Social Interaction Analysis** Several datasets have been developed to analyze human social interactions, many of which focus on conversational data or multimodal behaviors. For example, the MELD (Friends TV Series) Dataset (Poria et al., 2019) and the CMU Multimodal Opinion Sentiment and Emotion Intensity (CMU-MOSEI) dataset (Zadeh et al., 2018b) provide annotated multimodal data for studying emotions and sentiment in dialogue. More closely related to our work, the SocialIQA (Sap et al., 2019b) dataset introduced 38,000 multiple-choice questions derived from the ATOMIC (Sap et al., 2019a) knowledge graph, which is a large-scale graph of commonsense knowledge. These questions are aimed at testing models' understanding of social norms, intentions, and emotional responses in social scenarios. Furthermore, the Social-IQ Dataset (Zadeh et al., 2019a) was designed to evaluate social intelligence in AI with human-to-human interaction videos, including multimodal question-answer pairs that assess the ability to understand and respond to social situations effectively. A recent benchmark, SOCIAL GENOME (Mathur et al., 2025), evaluates multimodal models' ability to generate grounded social reasoning traces from videos, incorporating fine-grained cues and external knowledge. **Social Interaction Analysis with Language Models** Several previous studies have focused on the social reasoning capabilities of language-model social agents. Theory of Mind (ToM) has been tested through a variety of tasks, often times through measuring an LLM's ability to understand others' mental states using a series of reasoning-specific tasks (Ullman, 2023), or identifying social errors and understanding the perspectives of participants through faux pas tasks (Shapira et al., 2023). Frameworks such as COKE utilize contextual meanings of input entities to more accurately map knowledge graphs to be used in LLMs (Wu et al., 2023a), and the SOCIALIQA benchmark similarly provides a collection of commonsense questions with appropriate and inappropriate responses to be used in LLMs (Sap et al., 2019b). In studying emotional understanding, psychometric assessments have been developed to measure emotional understanding of an LLM based on a given scenario (Wang et al., 2023). Benchmarks such as Socratis measured emotional intelligence utilizing a repository of emotional reactions and appropriate scenarios (Deng et al., 2023).

## 6 CONCLUSION

We present the Social Human Robot Embodied Conversation (SHREC) Dataset, consisting of 400+ real-world interaction videos, 10,000+ annotations and eight new benchmark tasks spanning error detection, attribute reasoning, interaction flow, and rationale/correction inference, SHREC offers a critical resource to address unique socio-cognitive limitations of embodied agents. Through systematic evaluation of state-of-the-art LLMs and VLMs, we find that their overall performance falls far short from human-level, which underscores the limitations of current models in social reasoning. We envision SHREC as a foundation for advances in social reasoning for embodied social AI agents.

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

## A    EXPERIMENTAL SETUP

We evaluate each model on three separately sampled subsets per task. We then compute the mean performance (e.g., accuracy, F1) across these subsets and report ± one standard deviation to capture the variability due to sampling. All local experiments were conducted on an internal compute cluster equipped with NVIDIA RTX A6000 GPUs, each with 48GB of memory. A total of four GPUs were available, as confirmed by system diagnostics. For open-source models, we estimate GPU memory requirements based on parameter size: DeepSeek-R1-Distill-Qwen-32B ( 32B parameters) requires 64 GB VRAM, LLaMA-3.2 ( 8B) 16 GB, LLaMA-3.2-11B-Vision-Instruct ( 11B) 22 GB, Llava-Next-Llama3 ( 13B) 26 GB, InternVL2-8B ( 8B) 16 GB, MiniCPM-V 2.6 ( 2.6B) 5 GB, and Paligemma ( 3B) 6 GB. These models were run sequentially, with each task requiring approximately 2 hours per model per subset. With 8 tasks and 3 sampled subsets per task, this yields an estimated 48 GPU-hours per model. Proprietary models such as OpenAI's GPT-4o and Google's Gemini 1.5/2.0 were accessed via their respective APIs, so hardware specifications are not available. Nevertheless, the same 3-subset evaluation protocol was used, and the number of prompts per task was matched to ensure comparability. The total compute estimate for open-source models amounts to approximately 480 GPU-hours, with API-based models assumed to have undergone a similar number of queries.

The below describes the number of samples for each task used for our evaluation. Given that we utilize the overlapping and agreed samples for our tasks, in sum, we have $\sim$8040 available annotations for Tasks: (1-4) and (5-6):Social Error and Competence Detection, Error Detection, Social Attribute Identification, Multiple Attribute Presence Test, Correction and Rationale Reasoning. For the pre-condition and post-condition tasks, you are able to sample from the competent samples of the dataset, yielding in $\sim$ 4500 samples. However, one could construct the task with varying pre and post condition choices, resulting in even more combinations for testing.

### A.1    SOCIAL ERROR AND COMPETENCE DETECTION (EC)

We randomly sampled 3 seeds of $\sim$ 200 samples from the dataset ($\sim$600 samples in total), 300 samples are sourced from the empathic dataset and 300 samples are sourced from the wellness dataset. This yielded in samples corresponding to 29.5% samples in competency, 50% error, 20.5% none categories respectively.

### A.2    SOCIAL ATTRIBUTE IDENTIFICATION

Similarly, We randomly sampled 3 seeds of $\sim$70 samples from the dataset. This resulted with samples with the following label distribution: (1) Emotion: 15%, (2) Engagement: 34%, (3) Conversational Mechanics: 70%, (4) Knowledge State: 48%, (5) Intent of User: 48%, (6) Social Context and Relationships : 53%, (7) Social Norms and Routine: 30%. The proportion sums greater to 70% as multiple social attribute can co-occur in a given sample. Furthermore, as the task is conditional on whether or not the segment corresponded on social error or social competence. 66% of the samples in our evaluation set corresponds to a social error, 33% of the samples corresponds to social competence.

### A.3    IF-THEN REASONING

Similarly, We sampled 3 seeds of $\sim$70 samples from the dataset. Specifically, we acquired samples that were annotated as social competence, (i.e. 100% samples in competency, 0% error and 0% in None).

**Rationale and Correction Reasoning:** We sampled 3 seeds of $\sim$70 samples from the dataset. Specifically, we acquired samples that were annotated only as social error, (i.e. 0% samples in competency, 100% error and 0% in None).

# B  PROMPT EXAMPLES

PROMPT EXAMPLE 1: ERRORS AND COMPETENCE DETECTION (EMPATHIC DATASET)

> The social robotic agent is designed to be a social support companion that facilitates the exchange of emotionally relevant stories and employs narrative therapy techniques to enhance feelings of connection and belonging.
>
> You are given the **Images and Conversation History** between a social robotic agent (Jibo) and a participant.
>
> Answer the following questions about social interactions. Now, given the Images and Conversation History between the social agent (Jibo) and a participant, return whether the agent exhibits:
>
> 1. Social Competence
> 2. Social Error
> 3. None
>
> **Definitions:**
>
> - **Social Competence:** The ability to successfully conduct social interactions, which depends on the awareness and identification of social-emotional cues, the ability to process such cues, and the ability to decide on and express a normative response.
> - **Social Error:** Behaviors that violate social norms and degrade a user's perception of the robot's socio-affective competence, such as interrupting at inappropriate times.
> - **None:** Neither a social error nor competence is observed.
>
> **Answer the above from the following Images and Conversation History:**
> {Interaction Transcript}

PROMPT EXAMPLE 2: ERROR DETECTION (EMPATHIC DATASET)

> The social robotic agent is designed to be a social support companion that facilitates the exchange of emotionally relevant stories and employs narrative therapy techniques to enhance feelings of connection and belonging.
>
> You are given the **Images and Conversation History** between a social robotic agent (Jibo) and a participant.
>
> Answer the following questions about social interactions. Now, given the Images and Conversation History between the social agent (Jibo) and a participant, return whether the agent exhibits:
>
> 1. Social Error
> 2. None
>
> **Definitions:**
>
> - **Social Error:** Behaviors that violate social norms and degrade a user's perception of the robot's socio-affective competence, such as interrupting at inappropriate times.
> - **None:** Neither a social error nor competence is observed.
>
> **Answer the above from the following Images and Conversation History:**
> {Interaction Transcript}

PROMPT EXAMPLE 3: SOCIAL ATTRIBUTE IDENTIFICATION (EMPATHIC DATASET, COMPETENCE SAMPLE)

> The social robotic agent is designed to be a social support companion that facilitates the exchange of emotionally relevant stories and employs narrative therapy techniques to enhance feelings of connection and belonging.
> You are given the **Images and Conversation History** between a social robotic agent (Jibo) and a participant.
> The following interaction has been labeled as an instance of **Social Competence**.
> Select which of the following social attributes it is most related to:
>
> 1. Emotions: The ability to identify and interpret emotional expressions in oneself and others
> 2. Engagement: Observing and assessing levels of participation and interest
> 3. Conversational Mechanics: Understanding turn-taking, interruptions, and conversational flow
> 4. Knowledge State: Assessing what others know or believe in context
> 5. Intention: Inferring the goals or purposes behind others' actions or speech
> 6. Social Relationships: Understanding interpersonal dynamics and their context
> 7. Social Norms: Recognizing accepted behaviors and violations in social settings
>
> **Answer the above from the following Images and Conversation History:**
> {Interaction Transcript}

PROMPT EXAMPLE 4: MULTIPLE SOCIAL ATTRIBUTE PRESENCE (WELLNESS DATASET)

> The social robotic agent is designed to be a social positive psychology coach that delivers interactive positive psychology interventions and provides other useful skills to build rapport with college students.
> You are given the **Images and Conversation History** between a social robotic agent (Jibo) and a participant. The following interaction corresponds to a case of **Social Competence** (competent socio-affective behavior).
> Consider the following seven social attributes:
>
> 1. Emotions: The ability to identify and interpret emotional expressions in oneself and others
> 2. Engagement: Observing and assessing levels of participation and interest
> 3. Conversational Mechanics: Understanding turn-taking, interruptions, and conversational flow
> 4. Knowledge State: Assessing what others know or believe in context
> 5. Intention: Inferring the goals or purposes behind others' actions or speech
> 6. Social Relationships: Understanding interpersonal dynamics and their context
> 7. Social Norms: Recognizing accepted behaviors and violations in social settings
>
> **Task:** Based on the transcript, determine whether the agent's behavior involves *multiple social attributes*. Respond with `True` if the behavior demonstrates more than one social attribute. Respond with `False` if the behavior is based on only a single attribute.
> **Answer the above from the Conversation History:** {Interaction Transcript}

PROMPT EXAMPLE 5: IF-THEN REASONING: PRE-CONDITION (EMPATHIC DATASET)

---

The social robotic agent is designed to be a social support companion that facilitates the exchange of emotionally relevant stories and employs narrative therapy techniques to enhance feelings of connection and belonging.

You are given the **Images and Conversation History** between a social robotic agent (Jibo) and a participant.

Now, we provide what the social agent did in response to a user behavior. From the following ordered choices and aligned images (i.e., the first image corresponds to the first agent behavior (1)) of users' behaviors: (1), (2), (3), (4), (5), select which user's behavior was the appropriate **pre-condition** for the agent's action.

**Agent Behavior:**

> **Agent:** Taking a walk and getting some fresh air can definitely help clear the mind and recharge. It's great to hear that you took a moment for yourself. By the way, do you have a story from your journal that you'd like to share? I'm all yours. Great. I'll be listening.

**Context (Conversational History):**

1. **Participant:** Not yet. I'll tell you about that next week.

2. **Participant:** Let's see. Let's see.

3. **Participant:** Yes. Today I took a walk around the building that I work in. I took the stairs all the way down four floors, and then all the way back up so that I could recharge to get back to work.

4. **Participant:** I don't know if I could say a late. Well, yeah, I did, but I've just been staying up late because I get home late and, you know, I go to work late and so it's kind of just a little bit of a cycle. I was trying to finish this book and I did finish it, which is, you know, felt good about that. I started a new book and. Yeah, it's just that. Just, you know, getting laid and everything.

5. **Participant:** It left me sore, but it was worth it because it was fun and I got to hang out with my friend and her mom.

---

PROMPT EXAMPLE 6: IF-THEN REASONING: POST-CONDITION (EMPATHIC DATASET

---

The social robotic agent is designed to be a social support companion that facilitates the exchange of emotionally relevant stories and employs narrative therapy techniques to enhance feelings of connection and belonging.

You are given the **Images and Conversation History** between a social robotic agent (Jibo) and a participant.

Now, we provide the user's behavior. From the following ordered choices and aligned images (i.e., the first image corresponds to the first context (1)) of the social agent's behaviors: (1), (2), (3), (4), (5), select which agent's behavior was the appropriate **response** to the user's action.

**User Behavior:**

> **Participant:** Yes. Today I took a walk around the building that I work in. I took the stairs all the way down four floors, and then all the way back up so that I could recharge to get back to work.

**Context (Conversational History):**

1. **Agent:** That sounds exciting. Looking forward to hearing about it next week. So how did the rest of your day go?

2. **Agent:** Great. I'll be listening.

3. **Agent:** Taking a walk and getting some fresh air can definitely help clear the mind and recharge. It's great to hear that you took a moment for yourself. By the way, do

---

you have a story from your journal that you'd like to share? I'm all yours. Great. I'll be listening.

4. **Agent:** It sounds like you've been caught in a bit of a cycle lately, getting caught up in reading late into the night and then having it affect your sleep and daily routine. But congratulations on finishing a book. That's always an accomplishment. Worth celebrating. By the way, do you have any stories or thoughts you'd like to share? Maybe something interesting or exciting that happened recently or even a fictional story you've written in your journal? I'm all ears. Or. Well, text and would love to hear from you. Great. I'll be listening.

5. **Agent:** That's awesome. Spending time with friends and trying new activities can be really refreshing. Did you have a favorite move or technique that you learned during the kickboxing session?

PROMPT EXAMPLE 7: RATIONALE REASONING (EMPATHIC DATASET))

The social robotic agent is designed to be a social support companion that facilitates the exchange of emotionally relevant stories and employs narrative therapy techniques to enhance feelings of connection and belonging.

You are given the **Images and Conversation History** between a social robotic agent (Jibo) and a participant.

Now, we provide an interaction in which the social agent's behavior corresponds to a **Social Competence**. Select which is the correct reason behind the Social Competence.

**Answer format:** `Answer:  (1)`

**Conversation History:**

> **Participant:** I think recognizing the efforts that you're already making and how those are paying off and then looking at other options and talking to people who've been in similar situations would help me feel resilient and look forward.

> **Agent:** Absolutely. Recognizing your own efforts and acknowledging the progress you've made so far is a great way to stay resilient. Additionally, exploring other options and seeking guidance from people who have been through similar situations can provide valuable insights and inspiration. Keep up the positive mindset and keep moving forward. As our conversation comes to an end, I want to express my gratitude for sharing your personal stories and experiences. It has been a pleasure talking with you today. Remember, you are capable of navigating through any challenges that come your way. I look forward to our next conversation. Take care.

**Reasons:**

1. The robot responds to the user's sharing in an encouraging way.

2. The robot reflects on specific details and compares her story with similar ones.

3. The robot reflects on details the user shared and asks a follow-up that puts the user into the context of the story the robot shared.

4. The robot recalls details and asks a follow-up.

5. Expresses gratitude to the participant for them sharing their experiences.

PROMPT EXAMPLE 8: CORRECTION REASONING (WELLNESS DATASET)

---

The social robotic agent is designed to be a social positive psychology coach that delivers interactive positive psychology interventions and provides other useful skills to build rapport with college students.

You are given the **Images and Conversation History** between a social robotic agent (Jibo) and a participant.

Now, we provide an interaction in which the social agent's behavior corresponds to an **Error in socio-affective behavior**. From the following numerical choices, select which behavior Jibo (the social agent) should have done instead.

**Conversation History:**

> **Participant:** You're facing the wrong way.
>
> **Agent:** Did you say you were ready? Okay, let me know when you're ready to continue some other time.

**Behaviors:**

1. Jibo should've skipped the session, remembering doing this at least two other times in the past.

2. Jibo should've helped the user log his mood, which is a common thing to ask Jibo.

3. The robot should've understood that "I'm finished" means the user is ready, and should've responded quicker to this.

4. The robot should've continued where they left off.

5. Should have replied to the participant's question.

---

## C    CO-OCCURENCE: SUBJECTIVITY IN ATTRIBUTE IDENTIFICATION

Social attribute prediction in our dataset is inherently a multi-label problem and shaped by two key factors. First, attributes often co-occur within a single segment, for example when a conversational turn simultaneously involves errors in both conversational mechanics and knowledge state. Second, annotators can subjectively disagree on which attributes are present, reflecting perceiver-dependent interpretations of social constructs Searle (1998); Mathur et al. (2024). Indeed, in our dataset annotators disagree 8.7% of the time, and 14.9% of labels are non-overlapping across annotators. To analyze this further, we compared two subsets: agreed samples, where annotators converge on a common multi-label set, and disagreed samples, where annotators diverge and produce different but overlapping sets of labels. We then evaluated model predictions on these subsets with three metrics: F1 (per-label overlap), PM (partial match), and EM (exact match of the full set). We show these results in Figure 5.

We find that models score higher on disagreed samples for F1 and PM. This occurs because those metrics give credit for partially overlapping predictions: when multiple annotators mark many different attributes (on average 3.2 labels), the model is more likely to "hit" some of them by chance or partial alignment. In contrast, models score lower on disagreed samples with EM. EM requires the full set of labels to be predicted exactly, which is very unlikely when annotators themselves disagree. For agreed samples (on average 2.6 labels), EM scores are relatively higher, since the task is better defined and the ground truth is less subjective. These results suggest that PM can overestimate performance on subjective cases by rewarding partial overlaps even when no single ground-truth set exists. EM, while harsher, better discriminates between subjective and non-subjective cases, because it only succeeds when the model identifies the full set agreed upon by annotators. Thus, F1 provides a more conservative signal of whether models capture the precise constellation of social attributes that annotators consistently recognize.

## D    EXAMPLES OF ANNOTATIONS

**Social Competencies**:

**F1, PM, and EM Scores for Attribute Identification with Multiple Co-Occuring Attributes**

Figure 5: Our dataset offers annotations identifying errors, competencies with rationale and possible repair behaviors.

- Knowledge State: Participant 1: Video 0: 2:00-2:02; The robot remembers the user's name from earlier in the discussion and repeats it back to them, furthering the trust and potential relationship.

- Engagement: Participant 54: Video 19: 3:13-3:32; After thoroughly describing the next activity, the robot gauges the user's interest, specifically asks if they want to try the activity, and responds positively when they do.

- Emotions: Participant 34: Video 1: 5:21-5:29; The robot referred the user to resources based on how the user indicated they were feeling on the tablet.

- Social Norms: Participant 1: Video 0: 4:58-5:05; The robot thanks the user for their time at the end of the session

- Conversational Mechanics: Participant 1: Video 0: 0:27-0:35; The robot waits for the user to say her name before responding back.

- Intent: Participant 34: Video 1: 3:53-3:56; The robot previously told the user to say "I'm ready" when she's ready to continue. The robot continues correctly once it gets the phrase from the user.

- Social Context/Relationship: Participant 54: Video 19: 9:30-9:44; The robot fulfills its therapist role by encouraging steps the user can take out of the session, like practicing gratitude daily.

**Social Errors:**

- Knowledge State: Participant 54: Video 19: 2:00-2:43; After the user tells an extremely detailed, personal reflection, the robot only says "thanks for sharing" when the user finishes his story. It should have acknowledged the effort he put into his answer by repeating specific comments he made to show it listened.

- Engagement: Participant 34: Video 1: 2:46-3:10; The robot requests the user to perform a task on the tablet, but the user, unengaged, leaves the frame. The robot continues talking, but doesn't check in with the user.

- Emotions: Participant 34: Video 1: 0:00-0:09; The robot asked the user how things were going, and when the user responded with a neutral and unenthusiastic "ok," the robot responded positively with "sweet." The robot should've responded more sympathetically and/or taken it as an opportunity to learn more and figure out why the user is just "ok."

- Social Norms: Participant 54: Video 19: 0:34-0:45 ; The user is obviously annoyed and angered for the robot's errors; The robot doesn't address its mishaps, and instead tells the user to stay positive. It doesn't understand that it is the problem, nor that it should apologize.

| Social Attributes | Description and Annotators' Comments | Error / Competency |
|---|---|---|
| Recognizing Emotions | The robot asked the user how things were going, and when the user responded with a neutral and unenthusiastic "ok," the robot responded positively with "sweet." We agreed that this was an inappropriate response and that the robot should respond more sympathetically and/or take it as an opportunity to learn more and figure out why the user is just "ok." | Error |
| Recognizing Engagement | The robot did not notice nor check in with the user when they walked away from the camera and spent a considerable amount of time disengaged from the robot and session. Denison and I both noted this and believed the robot could have asked if the user was still there and/or set more of a time limit on responses to keep the user engaged and maintained more effective communication. The robot could have also asked questions regarding the lack of interest in the session and adapted from this | Error |
| Recognizing Conversational Mechanics | The robot doesn't know that the user is done telling her story and waits in silence, and the user doesn't know how to tell the robot it's over | Error |
| Understanding Knowledge State of the User and Self | Robot recognizes user's name and repeats it back to them; but we both marked it and included the same reasoning. Understanding knowledge state reflects that the robot remembers information about the user. | Competency |
| Understanding the Intent of the User | The robot tells the user to say "I'm ready" when she's ready to continue, and the robot continues correctly once it gets the phrase from the user. | Competency |
| Recognizing Social Context and Relationships | The robot introduces itself and describes how it will be "working with" the user and guiding them through social psychology topics. It almost comes across as a teacher or coworker as opposed to a therapist, but I think in terms of introducing itself and taking the lead, the robot fulfills its role. | Competency |
| Recognizing Social Norms | The robot ends the conversation by thanking the user for their time and suggesting they continue talking the next day. We think the general politeness makes this a social norms instance. | Competency |

Table 3: Examples Error and Competency Annotations from Dataset Jeong et al. (2023b). For more examples, refer to Appendix Sec. D

- Conversational Mechanics: Participant 34: Video 1: 4:14-4:37; The robot doesn't know that the user is done telling her story, and the user doesn't know how to tell the robot it's over; Also: 54: Video 19: 3:26-3:32 Delayed response of the robot

- Intent: Participant 34: Video 1: 0:51-1:19: The robot asks for the user's understanding of character strengths, and after the participant strongly says they know about them, the robot dives into an explanation to get them on the same page. However, the user is disinterested and begins wandering around the room, clearly not wanting to hear what they already know.

- Social Context/Relationship: Participant 54: Video 19: 0:05-0:30; The robot repeatedly does not respond to the user, which makes the user take charge of the conversation by saying "let's talk about wellness"; this flips the social relationship and removes the robot from its role as the facilitator

# E   RELATED WORKS TABLE

| Dataset / Framework | Source | Real World Interaction | Task | Correct/Incorrect or Error/Competency Labels | Modality | Duration | # of Samples | # of People |
|---|---|---|---|---|---|---|---|---|
| MELD | TV | ✗ | Sentiment Analysis and Emotion Recognition | ✗ | L+A+V | 30 Hrs / 1827 mins | 14,000 | ✗ |
| CMU-MOSEI | YouTube | ✗ | Sentiment Analysis and Emotion Recognition | ✗ | L+A+V | 65 Hrs / 3900 mins | 23,000 | 1000 |
| SocialIQA | CrowdSourcing | ✗ | Commonsense Inference | ✓ | L | ✗ | 38,000 | ✗ |
| Cicero | CrowdSourcing | ✗ | Commonsense Inference | ✓ | L | ✗ | 53,000 | ✗ |
| NormBank | CrowdSourcing | ✗ | Commonsense Inference (Social Norms) | ✓ | L | ✗ | 155,00 | ✗ |
| MoralExceptQA | Psychology Studies | ✗ | Moral Exception Question Answering | ✓ | L | ✗ | 148 | ✗ |
| CobraCorpus | AI + CrowdSourcing | ✗ | Pragmatic reasoning of Offensiveness | ✓ | L | ✗ | 32,000 | ✗ |
| CulturalNLI | AI + CrowdSourcing | ✗ | Cultural Context Inference | ✓ | L | ✗ | 2,700 | ✗ |
| SocialIQ | YouTube + CrowdSourcing | ✗ | Social Reasoning QA | ✓ | L+A+V | 21 Hrs / 1, 200 mins | 7,500 | ✗ |
| SOTOPIA | AI | ✗ | Simulation of social agents in social scenarios | ✓ | L | ✗ | ✗ | ✗ |
| **Ours** | Real-World Deployment | ✓ | Error/Competence Detection Social Attribute Reasoning Inferential Reasoning Explainability | ✓ | L+A+V | 58 Hrs / 3500 mins | 10,214 | 58 |

Table 4: L: Language, A: Audio, V: Video

While our benchmark includes 400 videos, consisting of 3600 minutes of real-world human–robot interaction footage, this scale is on par with or larger than many multimodal datasets in social interaction research (e.g., Social Genome Mathur et al. (2025): 280 minutes, Social-IQ Zadeh et al. (2019b): 1200 minutes, MEmoR Shen et al. (2020): 2800 minutes, CMU-MOSEI Zadeh et al. (2018a): 3900 minutes). We also refer the reviewer to Appendix D, where we provide a comparative table of related works for clarity. Importantly, what distinguishes our dataset is that it is collected from real-world deployments of physically embodied social robots, offering naturalistic, longitudinal interactions—an extremely scarce setting, with, to the best of our knowledge, no publicly available data at this scale with equally comprehensive annotations. While we agree that broader coverage across domains (e.g., workplace, clinical) would be valuable, SHREC represents a strong and novel first step for evaluating embodied social reasoning, and we hope it will be expanded upon by the community over time.

# F   ERROR IDENTIFICATION PER ATTRIBUTE

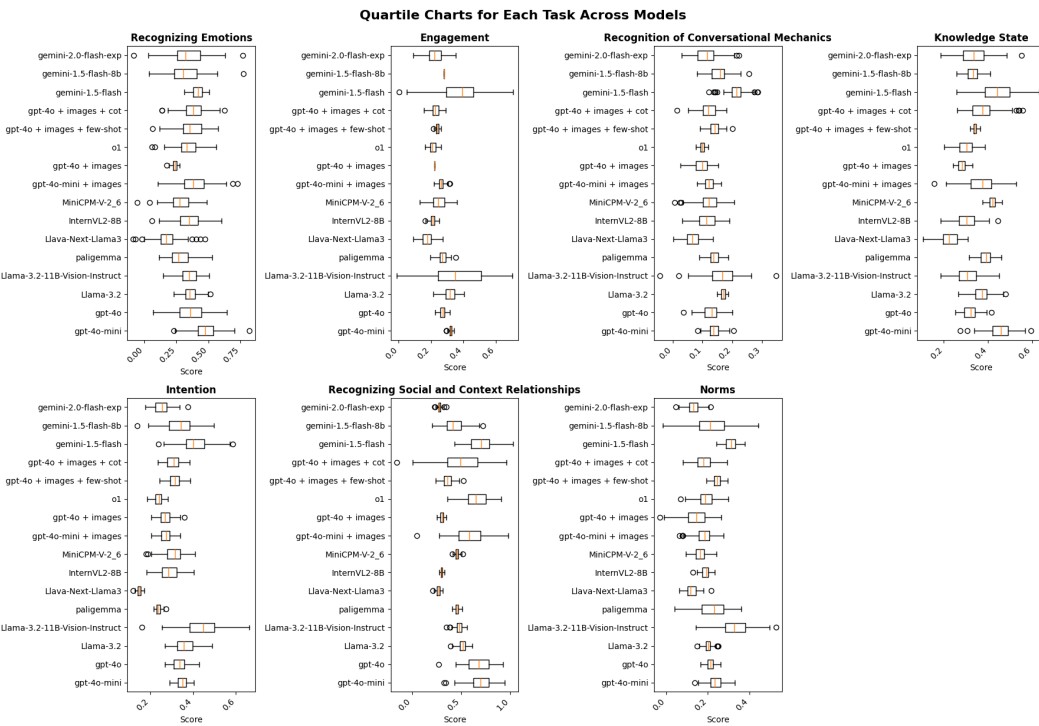

Figure 6: Error Per Attribute

We carry out further analysis to identify for which attribute the model is better at detecting errors. This highlights the usefulness of our benchmark, enabling analysis on when LMs fail, for which

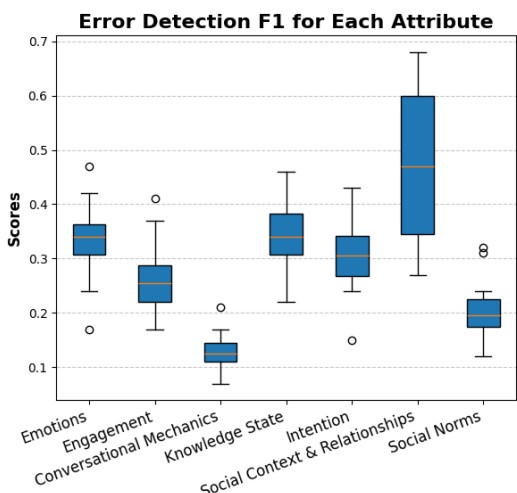

Figure 7: F1 scores on error detection for each attribute. Boxplot is constructed over the results of all 17 models.

attribute and why, as our dataset contains aligned comprehensive labels for every annotation. We report the F1 scores of error detection conditioned on the type of attribute the error is related to. While no models perform well on every single attribute, models tend to perform well on the social context and relationship category, whereas they perform poorly on conversational mechanics and social norms. This plot demonstrates strengths and weaknesses of current LMs and identifies pressing areas of research.

## G ATTRIBUTE IDENTIFICATION PER ATTRIBUTE

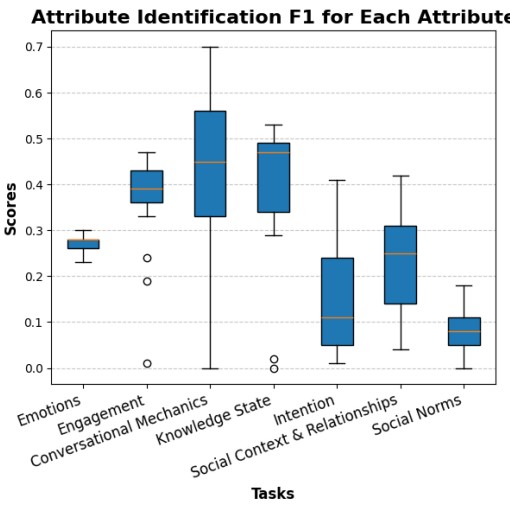

Figure 8: Attribute Identification F1 Per Attribute

In Fig. 8 we carry out further analysis to identify for which attributes model is good at identifying. This analysis showcases, given the true label for error or competency, whether the model is able to identify the related attribute. We find that given the error or competency label, still struggles to identify the correct attribute. More specifically, its performance on social norms, intention, social context and relationships are quite low. We find that certain models perform better than others in specific attributes GPT-4o variants excel in predicting attributes such as emotion, engagement,

knowledge state, and intention. In contrast, Llama-3.2 demonstrates strength in identifying social context and relationships, while Gemini shows better performance in conversational mechanics.

## H ANNOTATION PROCEDURE

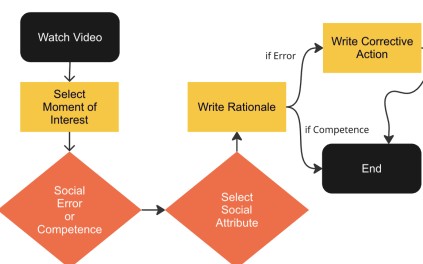

Figure 9: Annotation Procedure: Annotators watch the video, select moment of interest, then label for error or competence, then they select the social attribute and are asked to write the reason and alternate corrective behavior.

You are be given a video of a social robotic agent and you will be asked to annotate the agent's behavior where it exhibits social competence or an Social error. Here we share the definitions:

**Social Competence:** Social competence is the ability to successfully conduct social interactions, which depends on the awareness and identification of social-emotional cues, the ability to process such cues, and the ability to decide on and express a normative response to these cues.

**Social Error:** are errors that violate social norms and degrade a user's perception of a robot's Social competence, such as interrupting a user at an inappropriate time during a conversation

Simply put, Social competence refers to skillful social and affective behavior that is aligned to the desired and/or normal behaviors expcted by a user, thereby increasing trust, reliability, and overall perceived competence of the agent. Social error refers to a behavior exhibited by a robot that deviates from the desired or normal behaviors expected by a user, thereby degrading the overall perceived competence of the agent. As you may be able to tell, Social competence and error refers to behaviors that deviate or are aligned to the user's expectations. This specifically involves first recognizing, then responding appropriately to social and affective contexts. Here we share specific social attributes and definitions.

- **Emotions:** The ability to identify and interpret emotional expressions in oneself and others, allowing for empathetic responses and social awareness, e.g. recognizing that someone crying might mean they're sad Golan et al. (2006).
- **Engagement:** The skill to observe and assess levels of participation and involvement in social interactions, including cues that indicate interest or disinterest, e.g. continuing to tell a story when a listener is engaged Davis (1980).
- **Conversational Mechanics:** Understanding the structure and flow of conversations, including turn-taking, interruptions, and cues for when to speak or listen, e.g waiting for another person to finish speaking before speaking Fusaroli & Tylén (2016).
- **Knowledge State:** The ability to assess what others know or believe, as well as being aware of one's own knowledge in social situations, e.g. user talks about their dog, remembering that the user has a dog Baron-Cohen et al. (1999).
- **User Intention:** The capacity to infer the goals or purposes behind the actions and words of others, facilitating better responses in social interactions, e.g. when the user says "I'll be right back", indicates that the user will vacate and then return Dziobek et al. (2006).
- **Social Context and Relationships:** The ability to identify and understand the dynamics of social relationships and the context in which they occur, influencing behavior and expectations, e.g. knowing how to act in front of a close friend vs colleague at work Baron-Cohen et al. (1999).

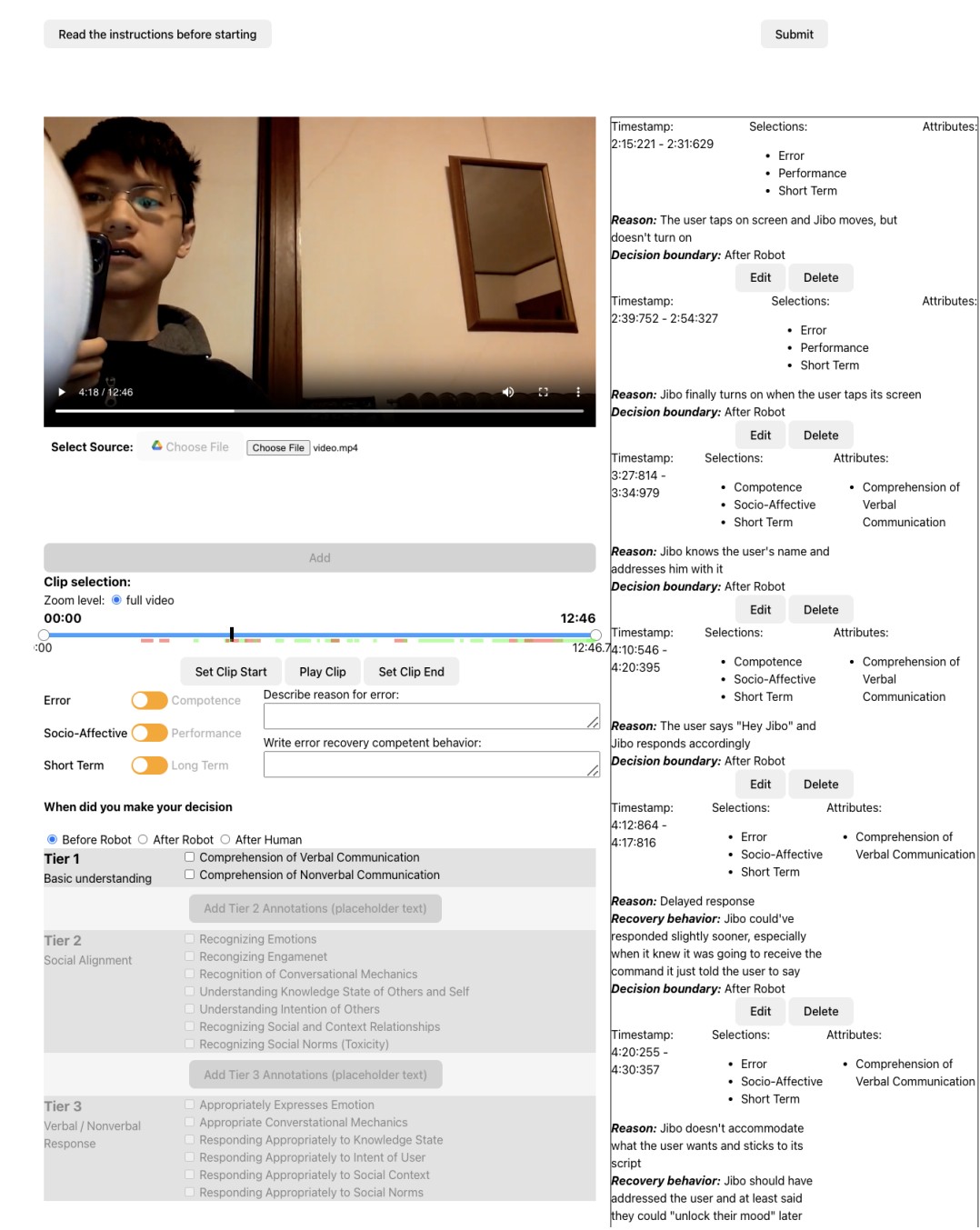

Figure 10: A screenshot of our annotation tool. Our tool enables the viewing of the video interaction, marking the moments of interest via sliders (after marking errors are in red, competencies are in green below the slidesrs, annotating for specific attributes). Typos were post-processed.

- **Social Norms and Routines:** The skill to identify accepted behaviors and attitudes within a social group, as well as recognizing negative or harmful interactions that violate these norms e.g. understanding that waving hands in the beginning of the interaction is a sign of a greeting Thoits (2004).

Then the annotators utilize the following annotation tool.

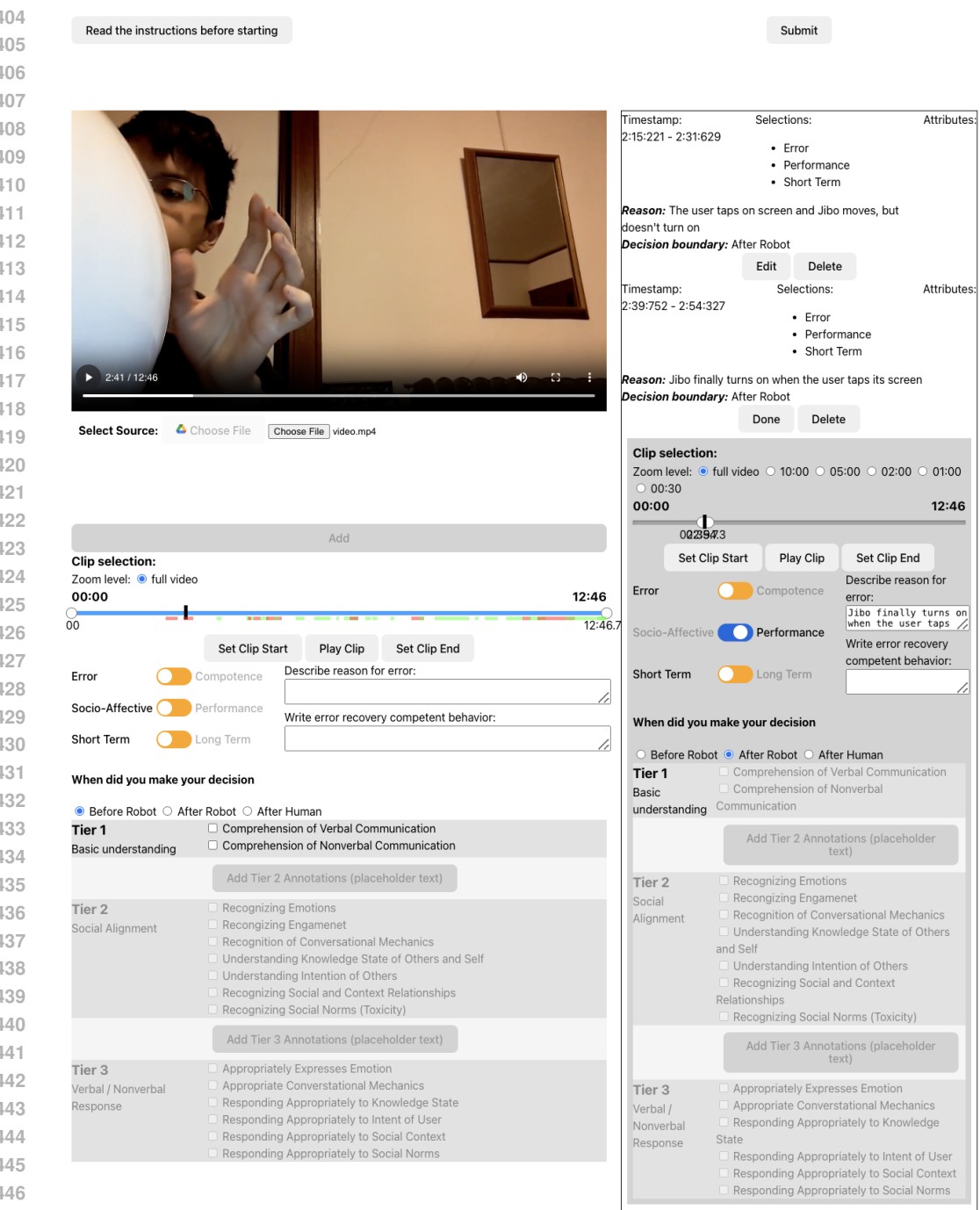

Figure 11: A screenshot of our internal annotation tool in the edit phase. Our tool flexibly allows the annotator to edit their previous annotations and look back at them. Typos were post-processed.

**Annotator Consistency** This protocol aligns with established best practices for achieving reliable subjective annotations Artstein & Poesio (2008). We further elaborate below. Each video was annotated independently by two trained annotators, who were free to identify segment boundaries for social errors and competencies. This flexible segmentation approach ensured that annotations captured the natural flow of interaction but introduced potential variability. Across the dataset, 85.1% of annotations overlapped, meaning that both annotators independently marked the same temporal segment. To ensure consistency, annotators underwent joint training with shared definitions and guidelines. Disagreements were initially addressed through collaborative review sessions on

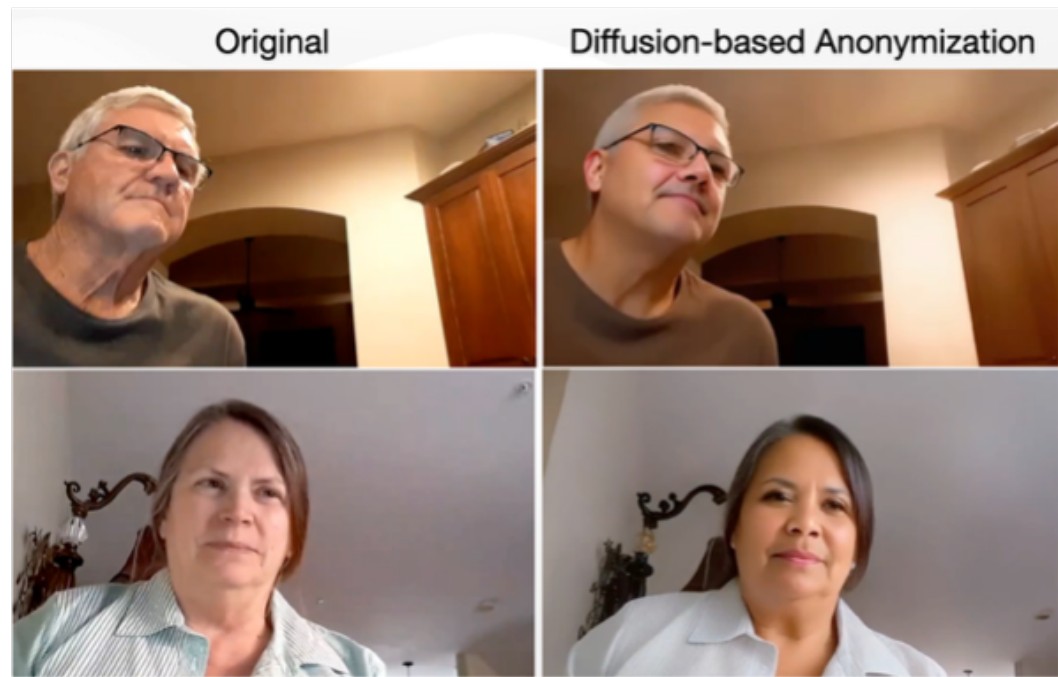

Figure 12: Original face (left) transformed into a fully synthetic version (right), preserving key social while ensuring privacy for responsible large-scale data release.

a held-out set, allowing annotators to align their interpretation of ambiguous or edge cases. After this calibration phase, annotators continued to annotate independently. For all non-overlapping or ambiguous annotations, the annotators reviewed each other's work and explicitly marked whether they agreed or disagreed with the alternate annotation. The remaining 8.7% of annotations were marked as persistent disagreements, reflecting segments where subjective interpretation of social behavior could not be reconciled. These were retained to preserve the richness and variability inherent to real-world human–robot interaction. This protocol balances rigor with the acknowledgment that social reasoning is inherently subjective. We view the residual disagreement not as noise, but as a valuable reflection of human subjective interpretations, we believe that it could also pose a meaningful challenge for AI models.

**Testing Instructions with Independent Annotators** We tested whether the guidelines would generalize to someone who had not participated in the initial discussions. A third annotator joined later in the process and was provided only the written instructions and example videos (without exposure to the earlier calibration sessions). When comparing this annotator's labels against the existing annotations, we observed a high agreement of 0.928 across the two original annotators' annotations, suggesting that the guidelines are clear and reproducible beyond the initial annotator pool.

## I   ANONYMIZATION ROBUSTNESS EXPERIMENT

To explore the impact of anonymization artifacts, particularly from FRESCO, we conducted a controlled experiment comparing vision-language model (VLM) performance across four video conditions:

- **Raw**: Unaltered, raw video

- **Diffusion**: FRESCO, Face replacement via text-guided diffusion (Yang, 2024)

- **Deepfake**: MobileFaceSwap replaces the face with random face images (Xu, 2022)

- **Cartoon**: VToonify stylized, animated face rendering (e.g., Disney style) (Yang, 2022)

We evaluated three VLMs—GPT-4o, Gemma-3, and LLaVA-Next—on a 3-way classification task (*Social Competence, Social Error, None*) using 10 balanced runs (15 samples each). Table 5 reports the average F1 and accuracy scores across conditions.

Table 5: Average F1 and accuracy scores (mean $\pm$ std) across anonymization conditions.

| Model | Raw | Diffusion (FRESCO) | Deepfake | Cartoon |
|---|---|---|---|---|
| GPT-4o | $0.462 \pm 0.083$ | $0.414 \pm 0.096$ | $0.513 \pm 0.058$ | $0.500 \pm 0.063$ |
| Gemma-3 | $0.194 \pm 0.050$ | $0.144 \pm 0.010$ | $0.196 \pm 0.045$ | $0.198 \pm 0.044$ |
| LLaVA-Next | $0.266 \pm 0.104$ | $0.309 \pm 0.119$ | $0.298 \pm 0.120$ | $0.260 \pm 0.075$ |

For GPT-4o, we observed a slight performance drop when using FRESCO compared to Raw and Deepfake videos. Interestingly, for LLaVA-Next, FRESCO slightly improved performance, possibly due to denoising or abstraction effects that helped the model focus on salient social cues.

Overall, Cartoon and Deepfake performed comparably or slightly better than Raw across models. Importantly, statistical analysis revealed no significant difference ($p > 0.05$) between Raw and any anonymized variant, including FRESCO.

These results suggest that even with visible visual artifacts, FRESCO does not significantly degrade model performance. We thank the reviewer for this suggestion and will include this experiment in the final version of the paper.

## J   LIMITATIONS

There are several limitations to our study that warrant discussion. First, the scope of social attributes we focus on—while grounded in existing frameworks—is not exhaustive. Social behavior is highly multifaceted, and other relevant competencies or error types may fall outside the seven social attributes we annotated. This limited coverage may reduce the generalizability of our findings to broader or less structured social scenarios.

Second, our annotation process relies on human judgment, which introduces potential subjectivity. While we implemented annotation guidelines and validation checks to improve consistency (see Appendix G), edge cases and disagreements (notably 8.7%) suggest that some annotations reflect annotator bias or ambiguity in interpretation. These effects may be compounded in complex, multi-label settings where social attributes co-occur, as shown in our attribute co-occurrence analysis (Appendix B).

Third, our dataset and experiments are geographically and culturally constrained, as all interactions were recorded in the United States. Social norms, conversational practices, and interpretations of robot behavior can vary significantly across cultures, limiting the cross-cultural robustness of both the benchmark and the model evaluations. Similarly, while we included a range of demographic backgrounds, some underrepresented subpopulations (e.g., elderly users, neurodivergent individuals) are insufficiently covered, which may impact the applicability of our findings in more diverse settings.

Fourth, there are strong modeling assumptions underlying many of the foundational models evaluated—such as independence between input modalities (in some architectures), idealized training corpora, and noise-free text or video representations. These assumptions may not hold in real-world settings involving ambiguous, noisy, or unstructured interaction data. Furthermore, our benchmark evaluates models at 1 Hz video sampling, which may miss subtle but socially relevant temporal dynamics.

Fifth, while we evaluate 17 modern LLMs and VLMs, each model is typically run with a single set of configurations and prompts, and we do not report variance across different random seeds or fine-tuning strategies. This limits the strength of claims about model generalization or robustness, especially given that some model performance differences may be attributable to configuration rather than inherent capability.

Sixth, due to the scarcity of large-scale real-world datasets in HRI, we prioritized collecting as much high-quality interaction data as possible across varied contexts. It is correct that the majority

of videos in SHREC primarily focus on the human participant, with the robot partially visible in many recordings. We acknowledge that this limits the dataset's ability to support systematic evaluation of the robot's full-body non-verbal behaviors or control policies. We will explicitly note this as a limitation in the final version and we also believe this highlights an important direction for future dataset collection: pairing human-centric recordings with synchronized robot-camera views or external wide-angle shots to better capture embodied robot behavior for modeling.

Our task formulations inevitably introduce sources of ambiguity that may affect both model and human performance. First, the Interaction Progression prompt relies on the notions of "pre-condition" and "post-condition" to indicate the likely motivation preceding an agent's action and the plausible continuation following a user's action. While we adopted these terms from prior NLP literature, they differ from their meaning in planning theory and may not perfectly capture the response–adjacency relation we intend. This choice of terminology and the conflated visual presentation of the two prompts could lead to confusion for both annotators and models. Moreover, because conversational responses often admit multiple reasonable antecedents or continuations (e.g., "yes" or "ok" may be appropriate in a wide range of contexts), the task may not have a uniquely correct answer in all cases. This introduces unavoidable variability in human judgments and may limit the interpretability of accuracy as a strict measure of correctness.

Furthermore, the construction of distractor items presents challenges. For Interaction Progression, multiple distractors may remain partially acceptable depending on how broadly one interprets conversational adjacency. For the rationale and correction tasks, our strategy of selecting distractors with different social-attribute annotations than the ground-truth response ensures surface-level distinctness, but may also reduce task difficulty. In particular, some incorrect rationales may differ sharply from the correct one, making the choice easier than intended. Conversely, there are cases where an action that remedies one type of social error could also plausibly remedy another, raising the possibility that more than one option could be considered correct. Thus, while our distractor selection method enforces consistency, it cannot guarantee the nuanced ambiguity of real-world social interactions.

Lastly, our current benchmark does not explicitly address fairness, bias, or privacy concerns beyond anonymization. While we use FRESCO for high-fidelity face stylization to preserve social signals and protect identity, potential demographic biases in model outputs (e.g., different error rates across user identities) remain unexplored. Future work should assess whether foundational models trained on SHREC exhibit biases in behavior interpretation or correction, particularly across identity markers such as gender, race, or age.

# K  BROADER IMPACTS

Our study addresses both the positive and negative societal impacts of our work, namely in how the SHREC dataset and benchmarks can pave the way for developing more socially adept robots that can be used to assist various populations in need. There is also a brief discussion of how possible bias may negatively impact perceptions of social behavior that the robot may be interpreting.

The creation of the SHREC dataset and its associated benchmarks stand to significantly advance the development of socially intelligent AI agents through grounding evaluation and training in actual human-robot interactions. By enabling a more fine-grained analysis and evaluation of social competencies and errors in real-world interactions, our study contributes a foundational resource for improving human-robot communication and interaction quality. Applications of this study could enhance assistive robotics, eldercare, education, and mental health support, where empathetic and socially appropriate behavior from robots can meaningfully improve user experience and well-being.

The societal deployment of socially adept AI agents does carry some notable risks. The improved social fluency in robots could lead to an overreliance on AI systems, particularly in vulnerable populations such as children or the elderly. This nuanced mimicry of human-like empathy by non-sentient machines may blur ethical lines in perceived agency and accountability. Additionally, there is a risk that biases in the interaction annotations may reinforce normative assumptions about social behavior, leading to potential exclusion or misinterpretation of diverse cultural or neurodivergent communication styles. These concerns emphasize the importance of ethical safeguards, continued human oversight, and more inclusive design practices as research in this area progresses.

**Environmental Impact** While the computational requirements for this work are modest compared to many large-scale AI projects, they still contribute to the overall carbon footprint of research. Based on an estimate of approximately 480 GPU-hours, we calculate an energy consumption of roughly 192 kWh (assuming a 0.4 kW power draw per GPU). Using a global average carbon intensity of 0.4 kg $CO_2$e/kWh, this corresponds to approximately 77 kg $CO_2$e, comparable to the emissions from driving an average passenger vehicle approximately 190 miles. We acknowledge that actual emissions will vary by GPU hardware, utilization, and local energy mix, and we encourage future work to report detailed compute and energy use. As part of our ongoing work, we are exploring strategies to minimize training runs, increase hardware efficiency, and utilize lower-carbon energy sources where possible.

**Safety and Security & Deception and Harassment** All three source studies that comprise SHREC (Wellness-Dorm, Wellness-Home, and Empathic++) were conducted under approved Institutional Review Board (IRB) protocols at our institutions, which explicitly address these issues. As part of the consent and briefing process, participants were clearly informed that the robot was not sentient, that its social behaviors were scripted or AI-generated, and that its role was to support a research study, not to provide professional, medical, or psychological advice.For example, consistent with ethical research practices,in the Wellness studies, we explicitly informed participants that the robot was not designed to assist with mental health emergencies and had limited perceptual capabilities (e.g., "Robot ears are different from human ears and I (Jibo) might have trouble understanding what you say"). Each robot station displayed a sticker with the National Suicide Prevention Lifeline hotline information. Pre-screening included the PHQ-9 depression questionnaire, and our protocol specified that anyone scoring above 20 (max score 27) would be excluded from participation; in practice, no participants met this exclusion threshold. These measures ensured that participants were aware of the robot's limitations and that appropriate safeguards were in place for any mental health concerns. Similarly, the onboarding protocol for the Empathic++ study explicitly informed participants that the robot lacked sentience and that its utterances were produced by OpenAI's GPT model.

**Data Quality and Representativeness & Discrimination, Bias, and Fairness** Our current dataset, drawn from three prior IRB-approved studies (Wellness-Dorm, Wellness-Home, and Empathic), reflects the participant pools of those original deployments. In the Wellness Jeong et al. (2020;?) studies (N=70), participants represented multiple racial and ethnic backgrounds: White (62.8%), Asian/Pacific Islander (28.5%), Black or African American (2.85%), Hispanic/Latino (2.85%), Native American (1.42%), and multi-racial (1.42%), with a mean age of 46 years and a standard deviation of 23 (Range: 18-83 years old) and consisted of consisted of 65.7% females, 28.6% males, and 5.7% others.. The Empathic (N=46) Shen et al. (2024) study participants ranged from 20-75 years old with a mean age of 36 and a standard deviation of 14.45, and contains 38.9% males and 61.1% females . While these samples provide rich, real-world interaction data, they are U.S.-centric and underrepresent certain age groups, particularly older adult populations. We will acknowledge this in the paper's Limitations section, but also stress that this work represents one of the largest, most richly annotated real-world social robot interaction datasets to date. Importantly, expanding diversity is an active and ongoing effort. Firstly, we have conducted prior studies with both older adults and children, and we plan to gradually integrate these interactions into the dataset to broaden its demographic coverage over time Park et al. (2017); Ostrowski et al. (2022). Secondly, our group leads the Jibo Community Social Robot Research Platform @Scale Park et al. (2024) initiative, we have converted previously commercialized Jibo robots into a shared, cloud-connected, community research platform. This infrastructure is designed to enable social robot living labs—multi-site, long-term deployments in diverse settings such as homes, schools, community centers, and senior living communities. By partnering with a broad network of researchers, including those embedded in underserved communities, we are already laying the groundwork for future SHREC expansions with broader cultural, geographic, and age diversity, ensuring the benchmark grows even more representative over time.

## L    ASSETS LICENSE

We benchmarked 17 large language and vision-language models, including both open-source and proprietary systems, on the SHREC benchmark. We also constructed our dataset using three prior

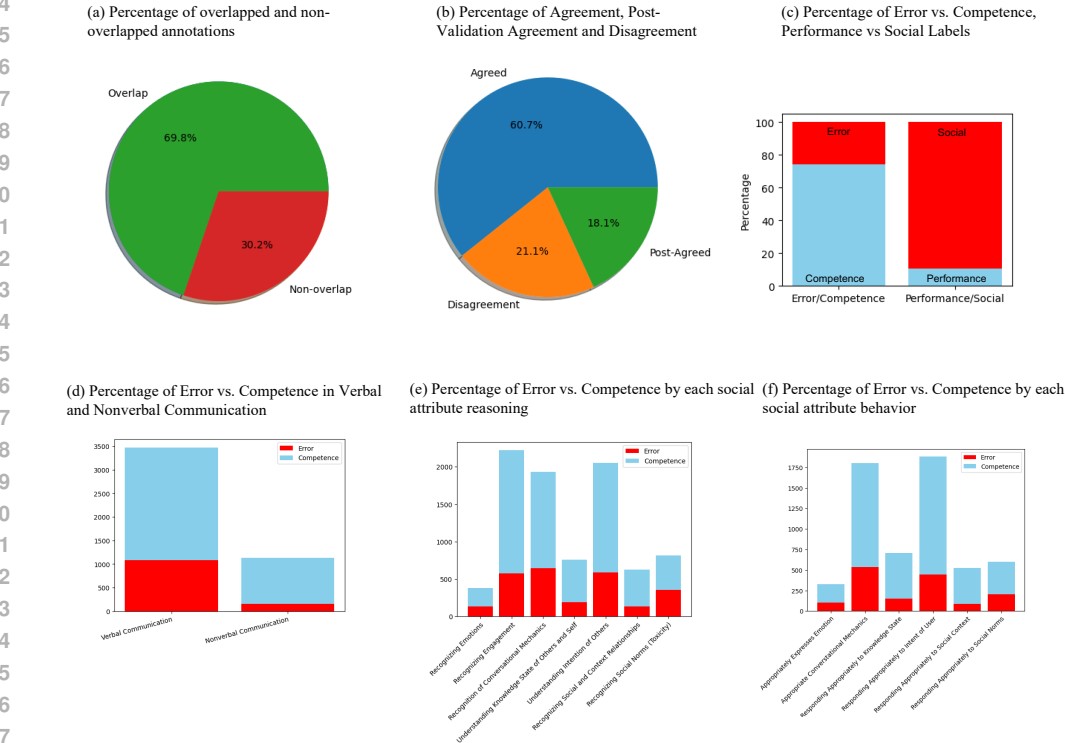

Figure 13: Wellness Jeong et al. (2023b) Dataset Statistics: We find that 69.8% of the dataset consists of overlapping annotations. Amongst the overlapping samples as shown in Figure B, we find an 78.1% overall agreement, where annotators agree on the error/competency and social/competency labels. In Figure C, again, we showcase the percentage of errors and competencies on the left and whether if they were related to social or performance. We find the majority being competencies relating to social dimensions. In the bottom row, we showcase plots regarding whether the error or competencies manifested in the perception, or the reasoning, or the behavior. In figure (d), we find that majority of the annotations marked by annotators belong in the verbal communication category. In figure (e) and (d), we find that most annotations belong in understanding or responding to (1) recognizing engagement, (2) conversational mechanics, (3) intent. . If we consider the competencies and errors separately, we find that annotators marked the most number of errors for conversational mechanics, intent and knowledge state and most number of competencies for engagement and social context.

human–robot interaction studies. Below we list each asset along with its creator and license or usage information, where available.

MODELS

- **GPT-4o**, **GPT-4o-mini**, **o1**
    - Creator: OpenAI
    - License: Accessed via API under OpenAI Terms of Use
- **Gemini 1.5**, **Gemini 2.0**
    - Creator: Google DeepMind
    - License: Accessed via API under Google AI Usage Terms
- **DeepSeek-R1-Distill-Qwen-32B**
    - Creator: DeepSeek AI
    - License: MIT License; Hugging Face link
- **LLaMA-3.2**

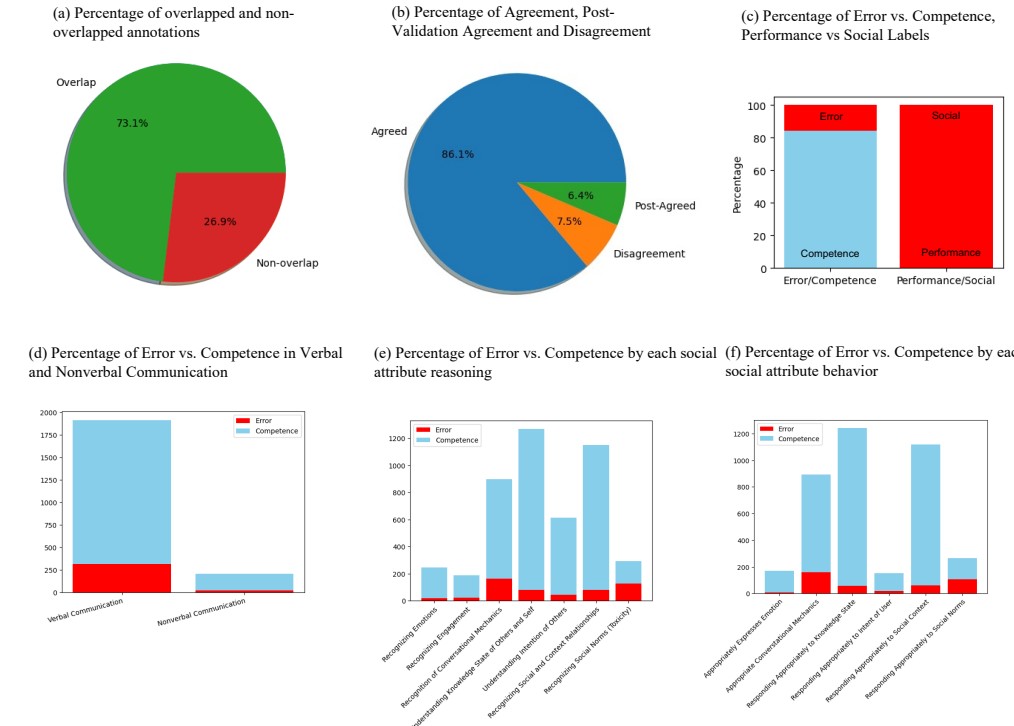

Figure 14: Empathic Shen et al. (2024) Dataset Statistics: We find that 73.1% of the dataset consists of overlapping annotations, where two annotators marked the sample . We refer the reader to Appendix REFER for the algorithm used to calculate overlaps. Amongst the overlapping samples as shown in Figure B, we find an 92.5% overall agreement, where annotators agree on the error/competency and social/competency labels. A random agreement would have been 25%. We go through a two-step procedure of this process, where in the first phase, the annotator simply annotates the video themselves. In the second step, the annotator looks at the other annotator's annotation and agrees whether or not this could be a possible interpretation. In Figure C, we showcase the percentage of errors and competencies on the left and whether if they were related to social or performance. We find the majority being competencies relating to social dimensions. In the bottom row, we showcase plots regarding whether the error or competencies manifested in the perception, or the reasoning, or the behavior. In figure (d), we find that majority of the annotations marked by annotators belong in the verbal communication category. In figure (e) and (d), we find that most annotations belong in understanding or responding to (1) knowledge state, (2) social relationships, (3) conversational mechanics. If we consider the competencies and errors separately, we find that annotators marked the most number of errors for conversational mechanics, and toxicity and most number of competencies for knowledge state and social context.

- – Creator: Meta AI
- – License: Meta Llama 3 Community License; License
- **LLaMA-3.2-11B-Vision-Instruct**
  - – Creator: Meta AI
  - – License: Same as above
- **LLaVA-Next-LLaMA3**
  - – Creator: LLaVA Team / Meta AI
  - – License: Built on LLaMA-3.2; inherits Meta's license
- **InternVL2-8B**
  - – Creator: OpenGVLab
  - – License: MIT License; Hugging Face link

- **MiniCPM-V 2.6**
  - Creator: OpenBMB
  - License: Apache 2.0 (code); MiniCPM Model License (weights); Hugging Face link
- **PaliGemma**
  - Creator: Google / Gemma
  - License: Apache 2.0 (code), CC-BY 4.0 (content); Docs

DATASETS

The SHREC dataset comprises real-world social interactions collected across three previously unpublished studies, now shared under a new IRB protocol. All data releases follow institutional guidelines for responsible dissemination.

- **Empathic++** Shen et al. (2024): A ChatGPT-powered empathic social robot facilitated emotionally meaningful conversations using narrative therapy techniques to promote connection and belonging.
- **Wellness-Dorm** Jeong et al. (2020): A socially assistive robot served as a positive psychology coach for college students in dormitories, delivering interventions on gratitude, strengths reflection, and goal-setting.
- **Wellness-Home** Jeong et al. (2023a): Robots were deployed in participants' homes under three conditions: assistant, coach, and companion, each offering varying degrees of social and functional support.

All datasets were anonymized using FRESCO, a diffusion-based video anonymization framework (MIT License), and manually filtered to remove personally identifiable information. The SHREC dataset is released via gated access on Hugging Face, with use governed by institutional IRB approvals.

## M  LLM USAGE

We used large language models (LLMs), specifically OpenAI's ChatGPT, as a writing and editing aid during the preparation of this manuscript. The LLM was employed to polish wording, improve clarity and flow, and rephrase certain sections for readability. The final text was reviewed and edited by the authors to ensure accuracy and originality.

