# OpenReview forum: "Social Human Robot Embodied Conversation (SHREC) Dataset: Benchmarking Foundational Models’ Social Reasoning"
_ICLR.cc/2026/Conference — Submitted to ICLR 2026_

### Official Review · Reviewer_TjUP · 2025-10-31

**Soundness:** 3
**Presentation:** 3
**Contribution:** 3
**Rating:** 6
**Confidence:** 3

**Summary:**

The paper introduces the Social Human Robot Embodied Conversation (SHREC) Dataset, containing 10,353 annotations from 403 interaction videos spanning over 3,500 minutes. With this dataset, the paper developed eight tasks to measure the social reasoning capability of foundational models. Benchmarking results over 17 FMs alongside human evaluations reveal substantial performance gaps, underscoring the difficulty and providing directions in developing socially intelligent AI.

**Strengths:**

- The paper contributes one of the largest real-world human–social robot interaction benchmark datasets.

- The annotation seems of high quality and valuable to the community to evaluate social intelligence.

- The benchmark is comprehensive, and the analysis is solid and insightful.

**Weaknesses:**

- The "embodied" conversation seems not well-justified in the paper, which should relate more to the physical body interactions or self-awareness, or egocentric perceptions. "Situated" might be a better term for the scope of the proposed dataset.

- The dataset is curated from three existing sources, with no new data source introduced.

**Questions:**

- What is the "we observe a significant gap between nascent open-source vision-language models (VLMs) and even text-only LMs," in line 428-429?

- How are nonverbal questions fed into pure text models?

- How many human hours and money are cost for this dataset annotation project?

**Details Of Ethics Concerns:**

There seems to be no report on annotator compensation information.

---

> ### Author Response · Authors · 2025-11-26
> **Embodied Conversation**
>
> > The "embodied" conversation seems not well-justified in the paper, which should relate more to the physical body interactions or self-awareness, or egocentric perceptions. "Situated" might be a better term for the scope of the proposed dataset.
>
> We would like to clarify that SHREC involves interactions with a physically present social robot, (an embodied agent), not a virtual agent, not a screen-based interface, and not a simulated VLM system. Participants engage with a robot that occupies space, has a physical body, moves its head and screen, and is co-located with the human during the interaction. Thus, the dataset captures genuine embodied human–robot interaction, not only situated or contextual dialogue.
>
> The physical presence of an embodied agent meaningfully alters human behavior, even when the robot has limited manipulation capabilities. A large body of HRI research demonstrates that co-present robots elicit qualitatively different patterns of gaze, engagement, trust, turn-taking, and spatial behavior compared to identical agents presented on screens (Kiesler & Goetz, 2002; Li, 2015). Humans modulate body orientation, interpersonal distance, leaning behavior, and timing of responses in ways that do not occur in webcam-based or text–image VLM interactions.
>
> Neuroscience research further supports this distinction: physically present robots activate mirror-neuron–related responses (Oberman et al., 2007), indicating that embodiment directly shapes how people perceive and interpret social behavior. These embodied effects arise from co-presence and are precisely the types of social signals, gaze shifts, proxemics, posture changes, repair behaviors that SHREC captures.
>
> For these reasons, we maintain that “embodied conversational interaction” is accurate: the robot’s physical embodiment fundamentally shapes the interaction dynamics recorded in SHREC.
>
> Roesler, Eileen, Dietrich Manzey, and Linda Onnasch. "A meta-analysis on the effectiveness of anthropomorphism in human-robot interaction." Science robotics 6.58 (2021).
> Kiesler, Sara, and Jennifer Goetz. "Mental models of robotic assistants." CHI'02 extended abstracts on Human Factors in Computing Systems. 2002.
> Oberman, L. M., McCleery, J. P., Ramachandran, V. S., & Pineda, J. A. (2007). EEG evidence for mirror neuron activity during the observation of human and robot actions: Toward an analysis of the human qualities of interactive robots. Neurocomputing, 70(13-15), 2194-2203.

---

> > ### Author Response · Authors · 2025-11-26
> > **How are nonverbal questions fed into pure text models?**
> >
> > > How are nonverbal questions fed into pure text models?
> >
> > We thank the reviewer for raising this point. In the main submission, nonverbal-related questions were simply provided to text-only models as-is (i.e., the prompt contained the question but no nonverbal information), since these models cannot ingest visual inputs. This ensured a clean comparison between text-only and vision–language models.
> >
> > However, motivated by the reviewer’s question, we additionally ran a new baseline in which all nonverbal cues are pre-extracted and converted into short textual descriptions, which are then appended to the text prompt before feeding into pure text LLMs.
> >
> > | Model                                    | Error/Comp/None Acc | Error/Comp/None F1 | Error Acc | Error F1 | Identification Acc | Identification F1 | Identification Partial    | Multiple Attr. Acc | Multiple Attr. F1 | Pre Acc    | Post Acc   | Rationale Acc | Correction Acc |
> > |------------------------------------------|----------------------|---------------------|-----------|----------|---------------------|-------------------|------------|---------------------|-------------------|------------|------------|----------------|----------------|
> > | **(L+V) gemini-2.5-pro**                 | 0.29±0.02           | 0.27±0.02          | 0.40±0.01 | 0.33±0.02 | 0.01±0.01          | 0.22±0.03         | 0.55±0.02 | 0.48±0.01          | 0.25±0.02        | 0.55±0.09 | 0.55±0.06 | 0.49±0.06      | 0.45±0.08      |
> > | **(L+V) Gemini-2.5-pro + desc. text**    | 0.29±0.04           | 0.28±0.03          | 0.40±0.03 | 0.35±0.03 | **0.04±0.01**          | **0.45±0.01**         | **0.82±0.02** | 0.49±0.01          | **0.34±0.01**        | 0.57±0.06 | **0.67±0.01** | **0.55±0.06**     | **0.48±0.07**  |
> >
> > As shown in the results below, adding descriptive text to Gemini-2.5-Pro improves the performance for many tasks, including social attribute identification, pre if-then reasoning, rationale and correction. We thank the reviewer for this suggestion and will include this variant for the camera-ready.

---

> > ### Author Response · Authors · 2025-11-26
> > **[Ethics] How many human hours and money are cost for this dataset annotation project?**
> >
> > > How many human hours and money are cost for this dataset annotation project?
> >
> > Thank you for raising this important point. Ethics, transparency, and responsible research practice are central priorities for our project, and we appreciate the reviewer highlighting the need to clearly report annotator compensation and effort. We employed two independent annotators for every video segment, covering approximately 3,600 minutes of interaction footage. Following our institution’s standard compensation rate of 17.70 dollars per hour, and given that annotation requires roughly 1.5× the real-time duration of each video, the annotation process amounted to approximately 180 human-hours and $3,200 in labor costs. We will make sure to include this information explicitly in the final version to improve clarity and ethical completeness.

---

> ### Author Response · Authors · 2025-11-26
> **No new data source introduced**
>
> > The dataset is curated from three existing sources, with no new data source introduced.
>
> We acknowledge the reviewer’s comment, but we want to clarify that although the underlying interaction recordings were collected in prior IRB-approved studies, **these data have never been released as a dataset** nor curated for machine-learning research. SHREC is not a repackaging of existing public datasets; rather, it is the first time these raw human–robot interaction videos have been organized, standardized, anonymized, annotated, and made available as a unified benchmark.

---

> ### Author Response · Authors · 2025-11-26
> **Performance Drop of VLMs from LLMs**
>
> > What is the "we observe a significant gap between nascent open-source vision-language models (VLMs) and even text-only LMs," in line 428-429?
>
> Recent studies show that adding vision to a strong LLM can degrade its original text-only reasoning abilities, particularly on out-of-distribution benchmarks.
>
> Dash et al. report that multimodal training reduces text-only win rates on m-ArenaHard, showing degradation compared to the original text model, even when text data is included during multi-modal tuning. Zhai et al., further show show that VLM’s multimodal fine-tuning causes VLMs to lose generalization on unseen datasets.
>
> These works show that paligemma and Llava-Next, which have LLM backbones like Gemma, Qwen or Llama, (1) can result in performance degradation, (2) especially for OOD tasks. In other words, our results reflect the same phenomenon: multimodal tuning can impair the LLM backbone, leading VLMs to perform worse than text-only baselines on SHREC’s challenging social-reasoning tasks.
>
> - Dash, Saurabh, et al. "Aya Vision: Advancing the Frontier of Multilingual Multimodality." arXiv preprint arXiv:2505.08751 (2025).
> - Zhai, Yuexiang, et al. "Investigating the catastrophic forgetting in multimodal large language models." arXiv preprint arXiv:2309.10313 (2023).

---

### Official Review · Reviewer_P5wr · 2025-11-01

**Soundness:** 2
**Presentation:** 3
**Contribution:** 2
**Rating:** 4
**Confidence:** 3

**Summary:**

The paper introduces SHREC, a real-world benchmark of ~400 human–robot interaction videos with10k+ annotations capturing social competencies, social errors, their underlying social attributes(seven categories), and rationales/corrections.

**Strengths:**

1. Eight well-defined tasks cover a full spectrum of social reasoning, from error and competence detection to rationale and correction generation.
2. The study benchmarks 17 leading LLMs/VLMs and shows a clear human–model performance gap, underscoring the benchmark’s challenge and value.
3. Independent annotations with ~91% agreement and multi-label structure ensure reliability while capturing the richness of real-world social interactions.

**Weaknesses:**

1. The paper’s so-called “real-world physical AI agents/robots” function more like voice-based chatbots equipped with camera access for visual input(1HZ), with interactions primarily limited to video and audio exchanges, rather than physical manipulation or embodied actions. Such a setup aligns more closely with a multimodal Vision-Language Model dataset rather than a genuinely embodied human-robot interaction dataset, where agents can physically engage with and influence their environment. Consequently, the proposed work appears to represent a multimodal VLM interaction benchmark rather than a truly embodied conversational framework. Similar multimodal datasets are already abundant in existing research, which somewhat limits the novelty of the contribution.
2. Although the paper claims to use FRESCO for high-fidelity face stylization to preserve social signals while protecting identity, Figures 10 and 11 still reveal discernible facial features, raising concerns about the adequacy of anonymization. Such visual cues could potentially allow re-identification, posing privacy risks. Moreover, it remains unclear whether all human participants appearing in the dataset provided informed consent for their images to be included and publicly shared. The absence of explicit ethical review or consent documentation may undermine the dataset’s credibility from an ethical and compliance standpoint.
3. All data are drawn from three prior deployments involving similar social robots and university-aged participants. The limited diversity of settings and demographics may restrict the generalizability of SHREC to other social contexts or robot morphologies.

**Questions:**

1. “Some scenarios that were too subjective to reach full consensus yielded some annotations in the Disagree category of 8.7%.” Such disagreement introduces additional subjectivity and ambiguity into the dataset, which may undermine the stability and interpretability of evaluation results.
2. In Table 2, the performance of GPT-4o with few-shot prompting is notably weaker than that of GPT-4o alone, which appears counterintuitive. Typically, few-shot examples are expected to enhance model performance on novel tasks or at least maintain comparable results. This anomalous outcome may indicate potential inconsistencies in the dataset design or task formulation.
3. Reporting Cohen’s κ or Krippendorff’s α could further strengthen dataset's reliability claims.

**Details Of Ethics Concerns:**

Although the paper claims to use FRESCO for high-fidelity face stylization to preserve social signals while protecting identity, Figures 10 and 11 still reveal discernible facial features, raising concerns about the adequacy of anonymization. Such visual cues could potentially allow re-identification, posing privacy risks. Moreover, it remains unclear whether all human participants appearing in the dataset provided informed consent for their images to be included and publicly shared. The absence of explicit ethical review or consent documentation may undermine the dataset’s credibility from an ethical and compliance standpoint.

---

> ### Author Response · Authors · 2025-11-26
> **Scope of Work: Real-world Physical Robots**
>
> > The paper’s so-called “real-world physical AI agents/robots” function more like voice-based chatbots equipped with camera access for visual input(1HZ), with interactions primarily limited to video and audio exchanges, rather than physical manipulation or embodied actions. Such a setup aligns more closely with a multimodal Vision-Language Model dataset rather than a genuinely embodied human-robot interaction dataset, where agents can physically engage with and influence their environment. Consequently, the proposed work appears to represent a multimodal VLM interaction benchmark rather than a truly embodied conversational framework. Similar multimodal datasets are already abundant in existing research, which somewhat limits the novelty of the contribution.
>
> We respectfully disagree with the characterization that our dataset is “just a multimodal VLM benchmark.” SHREC involves a real, physically embodied social robot (Jibo) situated in a shared environment with human participants. While our robot does not perform manipulation, it is a physically embodied agent, and the interactions in SHREC fundamentally differ from screen-based or webcam-based multimodal VLM datasets.
>
> First, physical presence itself meaningfully alters human social behavior, even when the robot’s action capabilities are limited. **A long line of work in human-robot interaction (HRI) shows that co-present robots evoke different attentional patterns, engagement dynamics, and communicative behaviors compared to virtual agents or disembodied chatbots.** For example, many studies demonstrate that humans attribute greater social presence, trust, and responsiveness to physically embodied robots than to identical agents presented on screens (Kiesler & Goetz (2002) and Li (2015)). These effects arise even when the robot’s motor abilities are constrained, underscoring that **physically present embodiment shapes interaction independently of manipulation capability**.
>
> Second, the neuroscience literature provides direct evidence that humans process robots differently when they are physically present. **Oberman et al. (2007) show that observing actions performed by physically present robots activates human mirror-neuron–related EEG signatures, approaching the responses elicited during human–human interaction** This supports our core claim: embodiment modulates social cognition, which directly impacts the kinds of social signals, repair behaviors, engagement cues, and conversational grounding that appear in our dataset.
>
> Third, unlike typical VLM datasets—often collected via crowd-sourcing, web videos, or screen-based dialog systems—SHREC captures in-person, co-located human–robot interactions, including gaze patterns, spatial configurations, proxemics, and embodied turn-taking behaviors that cannot be observed from 2D video-chat style data. Participants treat the agent as a physically co-present partner, orienting their bodies, shifting gaze, modulating speech rate, and leaning in/out—all behaviors that change when interacting with a real robot versus a multimodal chatbot.
>
> In short, physically present embodiment shapes interaction independently of manipulation capability. SHREC captures embodied, situated, physically co-present interaction dynamics that are absent from conventional multimodal VLM datasets. These embodied social signals are precisely what socially intelligent agents must learn to interpret. Therefore, SHREC goes beyond a vision–language benchmark and represents a genuinely embodied conversational interaction dataset grounded in established findings from HRI and social neuroscience.
>
> - Roesler, Eileen, Dietrich Manzey, and Linda Onnasch. "A meta-analysis on the effectiveness of anthropomorphism in human-robot interaction." Science robotics 6.58 (2021).
> - Kiesler, Sara, and Jennifer Goetz. "Mental models of robotic assistants." CHI'02 extended abstracts on Human Factors in Computing Systems. 2002.
> - Oberman, L. M., McCleery, J. P., Ramachandran, V. S., & Pineda, J. A. (2007). EEG evidence for mirror neuron activity during the observation of human and robot actions: Toward an analysis of the human qualities of interactive robots. Neurocomputing, 70(13-15), 2194-2203.

---

> > ### Author Response · Authors · 2025-11-26
> > **GPT-4o with Few-shot sampling performs worse than GPT-4o alone**
> >
> > > In Table 2, the performance of GPT-4o with few-shot prompting is notably weaker than that of GPT-4o alone, which appears counterintuitive. Typically, few-shot examples are expected to enhance model performance on novel tasks or at least maintain comparable results. This anomalous outcome may indicate potential inconsistencies in the dataset design or task formulation.
> >
> > Regarding the reviewer’s concern that GPT-4o with few-shot prompting underperforms the zero-shot setting: our initial few-shot setup used **randomly sampled in-context examples**, a common evaluation strategy (Brown et al., 2020), but this may not be well aligned with the structure of our multi-attribute social reasoning tasks.
> >
> > To investigate this, we conducted a follow-up experiment using a more targeted retrieval strategy: for each test instance, we selected **few-shot examples that collectively represent the seven social attributes defined in SHREC**. The goal is to provide the model with more informative demonstrations tailored to the multi-attribute nature of the task. Below is the improvements-bolded results, where each metric is bolded if it is strictly higher than the GPT-4o baseline for that column.
> >
> > | Model                                      | Error/Comp/None: Acc | Error/Comp/None: F1 | Error: Acc | Error: F1 | Attr. Identification: Acc | Attr. Identification: F1 | Attr. Identification: Partial        | Multi Attr. Acc | Multi Attr. F1 | Pre Acc       | Post Acc      | Rationale Acc | Correction Acc |
> > |--------------------------------------------|----------------------|--------------------|-----------|-----------|-------------|------------|----------------|------------------|-----------------|----------------|----------------|----------------|----------------|
> > | **(L+V) gpt-4o**                            | 0.26±0.03           | 0.25±0.03         | 0.50±0.02 | 0.40±0.02 | 0.01±0.02   | 0.21±0.02  | 0.51±0.04      | 0.49±0.01        | 0.36±0.01      | 0.64±0.05      | 0.66±0.05      | 0.39±0.08      | 0.39±0.08      |
> > | **(L+V) gpt-4o + Random Sampling Few-Shot** | 0.24±0.02           | 0.22±0.01         | 0.48±0.02 | 0.38±0.01 | 0.01±0.01   | 0.21±0.02  | **0.52±0.04**  | 0.49±0.01        | 0.35±0.02      | 0.62±0.04      | 0.64±0.03      | 0.39±0.04      | 0.38±0.04      |
> > | **(L+V) gpt-4o + Per-Attribute Few-Shot**   | **0.29±0.04**       | **0.28±0.03**     | 0.40±0.03 | 0.35±0.03 | **0.04±0.01** | **0.45±0.01** | **0.82±0.02** | 0.49±0.01        | 0.34±0.01      | 0.57±0.06      | **0.67±0.01** | **0.55±0.06** | **0.48±0.07** |
> >
> >
> > Our results show that randomly sampled few-shot exemplars produce similar or slightly worse performance than the zero-shot setting, confirming the reviewer’s observation. However, when few-shot examples are selected in a structured manner to cover all seven social attributes, performance improves substantially, outperforming other baselines in 8 of 13 metrics and yielding especially large gains in partial match for multi-attribute identification. These findings indicate that the weaker performance of GPT-4o under random few-shot prompting is not due to inconsistencies in the dataset but rather a mismatch between arbitrary examples and the task’s compositional structure. When few-shot examples reflect this structure, the model can better leverage SHREC’s fine-grained, multi-attribute labels, demonstrating the value of the benchmark’s design. We thank the reviewer for raising this point, as it directly motivated this more informative analysis.
> >
> > - Brown, Tom, et al. "Language models are few-shot learners." Advances in neural information processing systems 33 (2020): 1877-1901.

---

> > ### Author Response · Authors · 2025-11-26
> > **Cohen’s κ or Krippendorff’s α**
> >
> > > Reporting Cohen’s κ or Krippendorff’s α could further strengthen dataset's reliability claims.
> >
> > Thank you for the suggestion. In the revised manuscript, we will additionally report inter-annotator reliability using both Cohen’s κ and Krippendorff’s α. Across the full annotation set for error and competency labels, **we obtain Cohen’s κ = 0.7638 and Krippendorff’s α = 0.76392**, further supporting the consistency and robustness of our annotation protocol.

---

> ### Author Response · Authors · 2025-11-26
> **[Ethics] Adequacy of anonymization**
>
> > Although the paper claims to use FRESCO for high-fidelity face stylization to preserve social signals while protecting identity, Figures 10 and 11 still reveal discernible facial features, raising concerns about the adequacy of anonymization. Such visual cues could potentially allow re-identification, posing privacy risks. Moreover, it remains unclear whether all human participants appearing in the dataset provided informed consent for their images to be included and publicly shared. The absence of explicit ethical review or consent documentation may undermine the dataset’s credibility from an ethical and compliance standpoint.
>
> While the reviewer raises valid concerns about anonymization adequacy, we want to clarify an important point: as stated in Lines 089–092, the entire study received explicit IRB approval from our institution and all human participants appearing in the dataset provided informed consent for data collection and public release. More specifically, there were **two approved IRBs in place**, we acquired  (1) initial IRB with study design and informed consent for publication and dataset release (without anonymization) and (2) then we acquired a secondary IRB approval to further privacy and anonymization. **We only share the data of participants who provided explicit, informed, written consent.**
>
> Even beyond meeting these requirements of the first IRB, we went further to receive a second IRB to enhance privacy protection. All publicly shared data uses FRESCO-based face synthesis to fully replace identifiable facial regions with synthetic, non-reversible faces. Thus, only the anonymized, IRB-approved, participant-consented synthetic versions appear in the dataset, ensuring compliance and responsible large-scale data release.
>
> **Figures 10 and 11 intentionally show unanonymized examples (from a consented participant)** to illustrate pre-transformation visual quality; these are not the images included in the released dataset. As shown in Figure 12, the original real face (left) is transformed into a fully synthetic version (right) that preserves essential social signals while preventing re-identification.

---

> ### Author Response · Authors · 2025-11-26
> **Dataset Diversity**
>
> > All data are drawn from three prior deployments involving similar social robots and university-aged participants. The limited diversity of settings and demographics may restrict the generalizability of SHREC to other social contexts or robot morphologies.
>
> We agree with the reviewer that diversity is critical for socially grounded evaluation, and we emphasize that the SHREC dataset was deliberately constructed to reflect this diversity more comprehensively than typical HRI datasets. . As noted in Lines 1647–1668 (Appendix), SHREC is constructed from three prior IRB-approved real-world deployments—**Wellness-Dorm, Wellness-Home, and Empathic**, each involving extended, naturalistic interactions with physically embodied social robots. While these studies were conducted in the U.S., they include a substantially more heterogeneous participant pool than is typical for publicly available HRI datasets.
>
> Specifically, the Wellness Dorm and Home (N=70) include participants identifying as White (62.8%), Asian/Pacific Islander (28.5%), Black or African American (2.85%), Hispanic/Latino (2.85%), Native American (1.42%), and multi-racial (1.42%), with a broad age range (18–83 years, M=46, SD=23) and gender distribution of 65.7% female, 28.6% male, and 5.7% other. The Empathic study (N=46) further spans ages 20–75 (M=36, SD=14.45) with a gender composition of 61.1% female and 38.9% male. Hence, the **dataset is not limited to university-aged participants.** While the dataset remains U.S.-centric and underrepresents some demographics (e.g., older adults), it reflects realistic, ecologically valid social-robot interaction contexts drawn from multi-site, multi-study deployments.
>
> Importantly, **none of these datasets have previously been publicly released to the research community**. HRI datasets, especially those involving *actual physical robots in longitudinal naturalistic settings, are exceptionally rare* due to privacy, logistics, and IRB constraints. SHREC therefore meaningfully expands the diversity, scale, and ecological validity of publicly available HRI datasets.

---

> ### Author Response · Authors · 2025-11-26
> **Subjective Annotations**
>
> > “Some scenarios that were too subjective to reach full consensus yielded some annotations in the Disagree category of 8.7%.” Such disagreement introduces additional subjectivity and ambiguity into the dataset, which may undermine the stability and interpretability of evaluation results.
>
> Thank you for this thoughtful comment. While Section 2.2 and Appendix G of the manuscript provide a high-level summary of inter-annotator agreement (91.3%) and annotation protocol, we agree that additional detail would strengthen clarity. This protocol aligns with established best practices for achieving reliable subjective annotations (Artstein & Poesio, 2008).
>
> Importantly, we could have released a filtered version of SHREC containing only the 91.3% of segments with full consensus, still a substantial dataset with 9.5K annotations from 403 interaction videos spanning over 3,500 minutes. However, as we describe in Lines 1451–1499, for all cases in the remaining 8.7%, annotators carefully reviewed each other’s labels and explicitly indicated whether they endorsed or rejected the alternate interpretation. The disagreements that remained were not due to inattention or annotation errors; they reflect *persistent, irreducible subjectivity* in how humans interpret subtle social behaviors. We view the residual disagreement not as noise, but as a valuable reflection of human subjective interpretations, we believe that it could also pose a meaningful challenge for AI models.

---

### Official Review · Reviewer_3MN3 · 2025-11-01

**Soundness:** 4
**Presentation:** 3
**Contribution:** 4
**Rating:** 6
**Confidence:** 3

**Summary:**

The paper presents SHREC Dataset. It is a large-scale benchmark designed to evaluate the social reasoning abilities of foundational AI models in real-world human–robot interactions. It includes over 400 videos and 10,000 annotations capturing social errors, competencies, and rationales in tabletop robot conversations. Results show that current models significantly lag behind human performance, highlighting the need for more socially intelligent AI.

**Strengths:**

1. Very thorough experiments on current language models and vision language models, clearly indicating the usefulness of the dataset.

2. SHREC fills a critical gap by focusing on real-world human–robot interactions, which are underrepresented in existing social reasoning benchmarks that typically center on human–human dialogue.

**Weaknesses:**

1. Counterintuitively, the human evaluation is not significant high: up and down 70%. The paper explains how the tasks are difficult, but I think there should be some kind of verification of the dataset's quality and interpretability by conducting expert-level human check.

2. It seems from the images shown (without video demos I cannot say for sure) that the videos are not as descriptive as general videos provide. The relatively small gap between language only models and VLMs also indicates this.

**Questions:**

1. Do you have any means to justify your dataset's quality and interpretability?

2. If a descriptive text on top of the transcript is provided, containing e.g. the actions taken by each human in the video, how will the language only models perform and how will human perform?

---

> ### Author Response · Authors · 2025-11-26
> **Human Evaluation & Dataset's quality/interpretability**
>
> > Counterintuitively, the human evaluation is not significant high: up and down 70%. The paper explains how the tasks are difficult, but I think there should be some kind of verification of the dataset's quality and interpretability by conducting expert-level human check. Do you have any means to justify your dataset's quality and interpretability?
>
> Thank you for the thoughtful comment. We would like to clarify an important distinction between dataset-quality validation and the human-evaluation results shown in the benchmark.
>
> ### 1. Human Evaluation in the Benchmark Is a Different Procedure.
>
>  The reviewer’s ~70% human performance refers to a **separate experiment**, designed solely for **fair comparison with LLMs/VLMs**. In this setting, another human annotator received the same restricted inputs as the models (short transcript segment, limited frames, no full-video access). This setup is intentionally different from the full annotation process and does not assess annotation quality; rather, it provides an **upper bound on human performance under model-equivalent constraints, specifically the limited context window of the video and language tokens.**
>
> ### 2. Dataset Quality Verification (Inter-Annotator Reliability + Independent Annotator Test).
>
> As shown in Figure 2 and Table 1 of the submission, the underlying annotations exhibit strong reliability:
> - 91.3% agreement in error/competency annotations across overlapping samples,
> - As a part of this rebuttal, we further report, Cohen’s κ = 0.7638 and Krippendorff’s α = 0.76392, indicating substantial agreement.
> - Further, Appendix H in Line (1493~1499) reports an independent annotator check: a third independent annotator, who did not participate in the original annotation guideline design, reproduced the labels using only the written instructions and example videos: achieving 0.928 agreement with the original annotators. This demonstrates that the guidelines are clear, consistent, and reproducible beyond the initial annotator pool.
>
> The dataset’s quality and interpretability are validated through strong inter-annotator reliability and an independent annotator test. The lower human scores in Table 1, 2 reflect the difficulty of the benchmark tasks, not deficiencies in the dataset. Following the reviewer’s suggestion, we will also clarify this in the paper.

---

> ### Author Response · Authors · 2025-11-26
> **Frames vs Videos**
>
> > It seems from the images shown (without video demos I cannot say for sure) that the videos are not as descriptive as general videos provide. The relatively small gap between language only models and VLMs also indicates this.
>
> Thank you for the comment. The static frames in the paper do not fully convey the expressiveness of the interactions, and we encourage the reviewer to view the **full video demos in the supplementary materials**, where it is clear that the recordings contain rich social cues captured at high fidelity. To further validate this, we conduct an Anonymization Robustness Experiment (Appendix I) comparing model performance across Raw (unaltered video), Diffusion (FRESCO face replacement via text-guided diffusion), Deepfake (MobileFaceSwap), and Cartoon (VToonify stylized rendering) variants. Statistical tests show no significant difference in downstream task performance across any anonymized condition (p > 0.05), indicating that the anonymization pipeline preserves the relevant social information and that the videos contain sufficient nonverbal cues for models to exploit—if the models are capable of doing so.

---

> ### Author Response · Authors · 2025-11-26
> **LLM Performance with Descriptive Text Summary as Context**
>
> >  If a descriptive text on top of the transcript is provided, containing e.g. the actions taken by each human in the video, how will the language only models perform and how will human perform?
>
> To address whether providing a descriptive text summary of the video (e.g., the actions of each participant) would improve language-only performance, we ran an additional experiment augmenting the input with a concise, automatically generated description.
>
> | Model                                    | Error/Comp/None Acc | Error/Comp/None F1 | Error Acc | Error F1 | Identification Acc | Identification F1 | Identification Partial    | Multiple Attr. Acc | Multiple Attr. F1 | Pre Acc    | Post Acc   | Rationale Acc | Correction Acc |
> |------------------------------------------|----------------------|---------------------|-----------|----------|---------------------|-------------------|------------|---------------------|-------------------|------------|------------|----------------|----------------|
> | **(L+V) gemini-2.5-pro**                 | 0.29±0.02           | 0.27±0.02          | 0.40±0.01 | 0.33±0.02 | 0.01±0.01          | 0.22±0.03         | 0.55±0.02 | 0.48±0.01          | 0.25±0.02        | 0.55±0.09 | 0.55±0.06 | 0.49±0.06      | 0.45±0.08      |
> | **(L+V) Gemini-2.5-pro + desc. text**    | 0.29±0.04           | 0.28±0.03          | 0.40±0.03 | 0.35±0.03 | **0.04±0.01**          | **0.45±0.01**         | **0.82±0.02** | 0.49±0.01          | **0.34±0.01**        | 0.57±0.06 | **0.67±0.01** | **0.55±0.06**     | **0.48±0.07**  |
>
> As shown in the results below, adding descriptive text to Gemini-2.5-Pro improves the performance for many tasks, including social attribute identification, pre if-then reasoning, rationale and correction. We thank the reviewer for this suggestion and will include this variant for the camera-ready.

---

> > ### Comment · Reviewer_3MN3 · 2025-11-27
> >
> > Thank the authors for their response to my questions. It's good to see additional results on the language-only models. With that said, the fact that language-only models with video captioning perform better than video models in social scenarios slightly reduces the dataset's significance. Considering both parts, my score will not change.

---

### Official Review · Reviewer_FY2D · 2025-11-06

**Soundness:** 3
**Presentation:** 2
**Contribution:** 3
**Rating:** 4
**Confidence:** 3

**Summary:**

This paper investigates an important topic — enabling LLMs and VLMs to better understand subjective (social) content in videos. The authors define the problem through several sub-categories, collect and annotate the dataset themselves, and evaluate model performance under the designed scenarios. Based on these experiments, they draw conclusions about the models’ capabilities.

**Strengths:**

1. The problem addressed in this paper is both timely and valuable, with strong relevance to current model training practices and potential benefits for social good.
2. The authors have formulated the problem thoughtfully and designed their approach in a well-structured and deliberate manner.
3. The authors take appropriate care to preserve user privacy by re-generating faces using a generative model rather than using real user data directly.

**Weaknesses:**

1. There is a typo in Line 96: “n selecting” should be corrected.
2. The presentation quality can be improved, as it is currently difficult to follow. For example, the font size in the figures is too small, and each scenario description contains excessive text. The inclusion of clearer and more visually appealing figures would make the paper much easier to understand.
3. The model selection appears somewhat inconsistent and disorganized. For instance, the comparison table should be divided into two parts: one for open-source models and another for closed-source models. It is also unclear why the authors chose PaliGemma and Gemini-1.5-Flash but did not include Gemini-2.5-Pro. Since the chosen models are not the most recent ones, I am skeptical about the validity of the conclusions regarding the current capabilities of VLMs.

**Questions:**

1. In Line 99, the authors use “Hz.” Could the authors clarify the difference between “Hz” and “fps” in this context? From my experience, many VLMs also downsample videos to 1 fps for processing.
2. I am curious about how the authors handle ambiguity in interpretation. For example, in Line 145, the paper mentions that crying indicates sadness. However, crying can also occur in positive emotional contexts (e.g., an athlete crying with joy after winning an Olympic final). Do the authors’ methods account for such ambiguous cases?
3. Since the authors use a vision–language (V+L) model, I wonder whether audio information is also incorporated. If so, why not extend the framework to an omni-modal setting?

---

> ### Author Response · Authors · 2025-11-26
> **Model Selection**
>
> > The model selection appears somewhat inconsistent and disorganized. For instance, the comparison table should be divided into two parts: one for open-source models and another for closed-source models. It is also unclear why the authors chose PaliGemma and Gemini-1.5-Flash but did not include Gemini-2.5-Pro. Since the chosen models are not the most recent ones, I am skeptical about the validity of the conclusions regarding the current capabilities of VLMs.
>
> Thank you for your suggestion regarding splitting the comparison table for open-source and closed source models. We will reflect this change by the camera ready.
>
> Also, thank you for the helpful suggestion regarding model selection. At the time of writing, we aimed to include as many widely used and community-relevant models as were publicly available. We appreciate that the landscape evolves quickly, and **we designed our evaluation code (available in the supplementary materials) to be easily extensible, new models can be incorporated with only a few lines of CLI commands**. To this end, we run newer models like Gemini-2.5-Pro,and OpenAI GPT5.1 mini and report the results below:
>
> | Model                | Error/Comp/None Acc | Error/Comp/None F1 | Error Acc | Error F1 | Attr. Identification Acc | Attr. Identification F1 | Attr. Identification Partial | Multiple Attr. Acc | Multiple Attr. F1 | Pre Acc | Post Acc | Rationale Acc | Correction Acc |
> |----------------------|---------------------|---------------------|-----------|----------|---------------------|-------------------|---------|---------------------|--------------------|---------|----------|----------------|----------------|
> | **(L+V) gemini-2.5-pro** | 0.29±0.02          | 0.27±0.02          | 0.40±0.01 | 0.33±0.02 | 0.01±0.01          | 0.22±0.03         | 0.55±0.02 | 0.48±0.01          | 0.25±0.02         | 0.55±0.09 | 0.55±0.06 | 0.49±0.06       | 0.45±0.08       |
> | **(L) gpt-5.1-mini**   | 0.22±0.02          | 0.18±0.01          | 0.35±0.01 | 0.24±0.01 | 0.02±0.02          | 0.21±0.01         | 0.50±0.01 | 0.48±0.01          | 0.34±0.02         | 0.65±0.02 | 0.62±0.05 | 0.55±0.07       | 0.47±0.03       |
>
> We find that GPT-5.1-mini outperforms previous baselines on the Rational and Correction tasks. We will include these results in the main paper.

---

> ### Author Response · Authors · 2025-11-26
> **Hz vs. Fps**
>
> > In Line 99, the authors use “Hz.” Could the authors clarify the difference between “Hz” and “fps” in this context? From my experience, many VLMs also downsample videos to 1 fps for processing.
>
> Thank you for the question. *Yes, 1 Hz is equivalent to 1 fps.* In the current release, we provide videos at 1 fps to match common preprocessing pipelines in many VLMs, which typically downsample to 1 fps for efficient processing. We are also preparing a 30 fps version of the dataset that has undergone full diffusion-based anonymization; examples of these higher-frame-rate videos are included in the supplementary material. The 1 fps setting was chosen deliberately to support VLMs’ typical input requirements, while the forthcoming 30 fps release will further benefit methods that rely on higher temporal resolution. The 30 fps version will be disseminated before the conference.

---

> > ### Author Response · Authors · 2025-11-26
> > **Ambiguity in interpretation**
> >
> > > I am curious about how the authors handle ambiguity in interpretation. For example, in Line 145, the paper mentions that crying indicates sadness. However, crying can also occur in positive emotional contexts (e.g., an athlete crying with joy after winning an Olympic final). Do the authors’ methods account for such ambiguous cases?
> >
> > We appreciate the reviewer’s question regarding ambiguity in interpreting expressive behaviors such as crying. Our goal is precisely to evaluate **whether AI models can reach human-level interpretations given the same information humans have access to**. Ambiguity is a natural property of real social interaction, and our annotations explicitly reflect human judgments in these cases. For instance, given an ambiguous instance, human annotators can disambiguate these cases based on contextual cues such as language, facial affect, body posture, conversational content, and social context. These annotations (gold labels) represent the human interpretation of the event, not a deterministic rule such as “crying = sadness.” Furthermore, we have **two human annotators watch the video, therefore, we have agreed labels for interpreting ambiguous expressive behaviors.**
> >
> > Our tasks therefore do not assume that any single cue has a fixed meaning; instead, they assess whether models can integrate multimodal context to make socially intelligent inferences in the way humans do. Ultimately, resolving such ambiguity is a core challenge of social reasoning, and SHREC is designed to provide a standardized way to measure whether models can handle these cases as reliably as humans.

---

> > > ### Author Response · Authors · 2025-11-26
> > > **Omni-modal Setting**
> > >
> > > > Since the authors use a vision–language (V+L) model, I wonder whether audio information is also incorporated. If so, why not extend the framework to an omni-modal setting?
> > >
> > > Thank you for the thoughtful suggestion. We agree that extending the benchmark to a fully omni-modal (audio–visual–language) setting is an important future direction. In the present version of SHREC, we intentionally exclude raw audio for privacy and anonymization reasons. Our data consists of real human–robot interactions in which speakers may inadvertently reveal sensitive or identifying information. While our video anonymization pipeline reliably removes identifiable facial features, we have not yet reliably sourced an audio anonymization method that both (i) preserves socially relevant cues (e.g., prosody, emotional tone) and (ii) guarantees that identity, demographic traits, or sensitive content cannot be inferred.
> > >
> > > Rather than release partially anonymized audio that could pose re-identification risks, we chose to focus the initial social interaction benchmark on vision–language modeling—an area where rapid progress in VLMs and LLMs makes evaluation especially timely. We are actively exploring audio-preserving de-identification pipelines, and once we can ensure privacy without compromising social signal fidelity, we plan to extend SHREC to an omni-modal version.

---

> ### Author Response · Authors · 2025-11-26
> **Typos and Presentation**
>
> >There is a typo in Line 96: “n selecting” should be corrected.
>
> > The presentation quality can be improved, as it is currently difficult to follow. For example, the font size in the figures is too small, and each scenario description contains excessive text. The inclusion of clearer and more visually appealing figures would make the paper much easier to understand.
>
> Thank you for noting the typo in Line 96; we will correct it. We will also revise the figures to improve readability (e.g., larger fonts, less dense text) so the scenarios are easier to follow.

---

### Official Review · Reviewer_6CHN · 2025-11-06

**Soundness:** 3
**Presentation:** 3
**Contribution:** 3
**Rating:** 6
**Confidence:** 3

**Summary:**

This paper introduces SHREC, a new benchmark dataset designed to evaluate social reasoning in embodied AI agents, particularly social robots engaged in human–robot interaction (HRI). The dataset consists of 400+ real-world videos (~3,500 minutes) and 10,000+ human annotations, capturing social errors, competencies, rationales, and corrective actions across verbal and non-verbal channels.

It defines eight tasks probing four key aspects of social reasoning:

1.	Error and Competence Detection

2.	Social Attribute Identification

3.	Interaction Flow Reasoning (Pre/Post-condition inference)

4.	Rationale and Correction Generation

The authors benchmark 17 state-of-the-art LLMs and VLMs (including GPT-4o, Gemini, LLaMA, o1, etc.) and compare them with human baselines. Results show a persistent gap between model and human performance, highlighting social reasoning as a distinct and underexplored capability for foundation models.

**Strengths:**

1.	SHREC addresses a gap in multimodal social reasoning evaluation. Most prior work (e.g., Social-IQ, SocialIQA, MELD, ATOMIC) involves human–human data, whereas SHREC uniquely targets real-world human–robot interactions, which involve distinctive failure modes like delayed reactions or incorrect affective mirroring.
2.	Real, embodied setting – The dataset uses physically embodied tabletop social robots rather than simulated or text-based scenarios. The focus on embodied social reasoning makes it a valuable step toward practical social AI evaluation.
3.	Comprehensive task suite – The eight tasks cover multiple reasoning levels: detection (Error/Comp./None), attribution (social attributes), sequential reasoning (pre-/post-condition), and prescriptive inference (rationale and correction). This provides both diagnostic (which aspect fails) and generative (what to fix) signals for model improvement.
4.	Systematic benchmarking – Evaluating 17 models under consistent multimodal settings (text-only and vision+language) with human baselines strengthens the empirical value. By evaluating both chain-of-thought (CoT) and few-shot variants, the authors effectively probe how explicit reasoning and contextual examples influence models’ ability to infer social context and intent.
5.	Attention to privacy and realism – The use of FRESCO for video anonymization while maintaining spatiotemporal cues (e.g., gaze, affect) reflects a well-considered approach to preserving critical social cues while ensuring participant privacy.
6.	The 91% inter-annotator agreement reflects that annotators consistently recognized and categorized nuanced social behaviors, supporting the reliability of the dataset’s labeling process.

**Weaknesses:**

1. **Unclear modality dependence** – It’s often unclear which tasks rely primarily on textual vs visual cues.
- For example, “Social Error, Competence, None Detection” seems largely text-driven, while “Social Attribute Identification” might require multimodal inputs (e.g., gaze, engagement). However, the paper doesn’t quantify modality contribution or ablate verbal-only vs visual-only performance.
- Similarly, some VLMs (e.g., paligemma, Llava-Next) perform worse than text-only models, which warrants a deeper discussion of why multimodality doesn’t help.

2. **Fine-tuning ambiguity** – It is unclear whether the models are zero-shot, few-shot, or fine-tuned for SHREC tasks.
- While Table 1 shows “few-shot” and “CoT” variants, it’s not stated whether any task-specific fine-tuning was performed or whether prompts were unified across models.
- Clarifying this would affect fairness and reproducibility of comparisons.


3. **Task redundancy/confusion** – The difference between Social Error, Competence, None Detection and Social Error Detection tasks is minor and not clearly motivated. The latter seems to be a subset of the former (binary vs ternary classification. It’s unclear if both are needed.

**Questions:**

- **Task modality**: Which tasks require video (vision) versus text (transcript) input? Could the authors provide per-channel ablation (vision-only vs text-only) to quantify the contribution of each?

- **Fine-tuning**: Were all models evaluated zero-shot, or was there any fine-tuning or prompt adaptation on SHREC? I wonder if finetuning on the dataset helps with any task.

- **Task redundancy**: What distinct insights are gained from both “Social Error, Competence, None Detection” and “Social Error Detection”?

---

> ### Author Response · Authors · 2025-11-26
> **Unclear modality dependence**
>
> > **Unclear modality dependence** – It’s often unclear which tasks rely primarily on textual vs visual cues. For example, “Social Error, Competence, None Detection” seems largely text-driven, while “Social Attribute Identification” might require multimodal inputs (e.g., gaze, engagement). However, the paper doesn’t quantify modality contribution or ablate verbal-only vs visual-only performance.
>
> We thank the reviewer for this question. Following reviewer’s suggestions, further analysis reveals that reliance on verbal vs. non-verbal cues is not uniform but varies substantially across different social attributes, a result we believe is both surprising and informative.
>
> Most tasks rely primarily on textual (verbal) cues (majority of the annotations come from the verbal cues 76.7%) as indicated by our Figure 2(b), where annotators marked whether the errors and competencies came from verbal channel, or the non-verbal channel when considering the source (*where the annotators thought about what modality the annotation was related to*). We **provide such labels for use for future use** for the community.
>
> As per your suggestion, we investigate, for each task, the ratio of each source modality.
> - Detection: Verbal: 0.81 Non-Verbal: 0.19
> - Detection Error Only:  Verbal: 0.76 Non-Verbal: 0.24
> - Post:  Verbal: 0.76 Non-Verbal: 0.24
> - Pre:  Verbal: 0.76 Visual: 0.24
> - Rationale:  Verbal: 0.79 Non-Verbal: 0.21
> - Corrective Action:   Verbal: 0.92 Non-Verbal: 0.08
> - Attribute Identification:  Verbal: 0.82 Non-Verbal: 0.18
> - Attribute Multiple:  Verbal: 0.76 Non-Verbal: 0.14
>
> The distribution reflect the textual nature of the tasks, and that ground truth human annotators relied more on verbal cues, which is not surprising as in many cases the nonverbal information is complementary to the verbal information.
>
> However, inspired by the reviewer’s comment, interestingly, the reliance on textual and verbal cues actually is more dependent on the type of social attribute being studied. (Note that the verbal and nonverbal proportions may not sum to 1, as both nonverbal and verbal social signals could have contributed to the annotation.) We find that Engagement, Conversational Mechanic and Intention of Others relies more on non-verbal cues compared to other attributes.
>
> - Knowledge State of Others and Self: Verbal: 97.1%, Non-Verbal: 4.5%
> - Social Context and Relationships: Verbal: 94.2%, Non-Verbal: 11.5%
> - Social Norms: Verbal: 92.2%, Non-Verbal: 18.9%
> - Recognition of Conversational Mechanics: Verbal: 89.5%, Non-Verbal: 29.3%
> - Intention of Others: Verbal: 77.8%, Non-Verbal: 33.5%
> - Emotions: Verbal: 98.2%, Non-Verbal: 8.4%
> - Engagement: Verbal: 59.1%, Non-Verbal: 59.6%

---

> ### Author Response · Authors · 2025-11-26
> **Discussion on why VLMs (e.g., paligemma, Llava-Next) perform worse than text-only models**
>
> > Similarly, some VLMs (e.g., paligemma, Llava-Next) perform worse than text-only models, which warrants a deeper discussion of why multimodality doesn’t help.
>
> Recent studies show that adding vision to a strong LLM can degrade its original text-only reasoning abilities, particularly on out-of-distribution benchmarks.
>
> Dash et al. report that multimodal training reduces text-only win rates on m-ArenaHard, showing degradation compared to the original text model, even when text data is included during multi-modal tuning. Zhai et al., further show that VLM’s multimodal fine-tuning causes VLMs to lose generalization on unseen datasets.
>
> These works show that paligemma and Llava-Next, which have LLM backbones like Gemma, Qwen or Llama, (1) can result in performance degradation, (2) especially for OOD tasks. In other words, our results reflect the same phenomenon: multimodal tuning can impair the LLM backbone, leading VLMs to perform worse than text-only baselines on SHREC’s challenging social-reasoning tasks.
>
> - Dash, Saurabh, et al. "Aya Vision: Advancing the Frontier of Multilingual Multimodality." arXiv preprint arXiv:2505.08751 (2025).
> - Zhai, Yuexiang, et al. "Investigating the catastrophic forgetting in multimodal large language models." arXiv preprint arXiv:2309.10313 (2023).

---

> ### Author Response · Authors · 2025-11-26
> **Fine-tuning ambiguity**
>
> > Fine-tuning ambiguity – It is unclear whether the models are zero-shot, few-shot, or fine-tuned for SHREC tasks. While Table 1 shows “few-shot” and “CoT” variants, it’s not stated whether any task-specific fine-tuning was performed or whether prompts were unified across models. Clarifying this would affect fairness and reproducibility of comparisons.
>
> All models in our benchmark were evaluated without any task-specific fine-tuning on SHREC. Our main results use strictly zero-shot evaluation to ensure comparability. *Specifically for GPT-4o*, we additionally ran limited few-shot and chain-of-thought (CoT) prompting experiments as inference-time prompting only (i.e., no gradient updates or model adaptation). These exploratory runs provided small, preliminary gains but did not change the overall conclusions.
>
> This setup is intentional:  while supervised fine-tuning is possible in future research, we initially release SHREC as a testbed. **To ensure fair, reproducible comparisons across proprietary and open-source models, all models received the same unified prompts per task, with the same formatting, instructions, and answer choices.** We agree that fine-tuning could potentially improve performance, but exploring training strategies is outside the scope of this benchmark paper; our goal is to measure out-of-the-box social reasoning ability rather than train specialized models on SHREC.

---

> ### Author Response · Authors · 2025-11-26
> **Task redundancy/confusion**
>
> > Task redundancy/confusion – The difference between Social Error, Competence, None Detection and Social Error Detection tasks is minor and not clearly motivated. The latter seems to be a subset of the former (binary vs ternary classification. It’s unclear if both are needed.
>
> While the ternary Social Error / Competence / None task and the binary Social Error Detection task are related, they serve distinct evaluation purposes. The ternary task measures whether a model can distinguish not only inappropriate behavior but also socially competent actions, reflecting a higher level of social understanding. The binary task, in contrast, isolates the foundational ability to detect the presence of a social error, which is the first decision point in safety-critical interactions. This separation mirrors how reward models are used in RLHF, where Bradley–Terry–Luce–style preference training only requires reliably identifying better versus worse behavior. Our results show that while current models struggle with the full ternary categorization, they perform substantially better on the binary distinction. This makes the binary task a meaningful intermediate milestone: models that cannot yet identify competence can still help prevent socially inappropriate actions. Thus, the two tasks are not redundant. The ternary task evaluates higher-order social understanding, while the binary task assesses whether models have reached the prerequisite detection capability necessary for RLHF-based alignment pipelines.

---

### Meta-Review · Area_Chair_7jJY · 2026-01-06

**Summary:**

This work received a mix of weakly negative and positive reviews initially. Post-rebuttal, one reviewed option to maintain their positive-learning score, while I would hazard that the other reviewers would maintain or improve, but unlikely that any would bump their scores to 8. In other words, quantitatively and qualitatively, regard for this work is still lukewarm.

While the authors partially or fully addressed most concerns in their rebuttals, some remaining niggling concerns (not necessarily major or fatal) are about the value of the dataset (being part of already-published work, although not released) and whether the work is truly embodied. The authors argue that the data has been not released before, and their contributions are about curation, annotations, etc. Nonetheless, this does reduce the value/significance of this dataset paper. Regarding the latter, I buy their argument that the data was from physical robots, even if there were no manipulation tasks (being a social interaction dataset), but then the value of embodiment is also reduced somehwat if there is no physical manipulation.

Overall, unfortunately for a high-selective venue like ICLR, the reviewers were lukewarm even if major technical concerns were somewhat or adequately addressed. For a dataset paper on a topic such as social embodied conversation, perhaps a subsequent submisssion to a robotics venue (particularly social robotics, e.g. HRI) may be better received -- and certainly those other communities/audiences may find this work to be a more valuable and salient contribution.

**Reviewer Concerns:**

Pls see above.

**Reviewer Scores:**

-- 3MN3 was explicit about maintaining score at 6.
-- TjUP and 6CHN: I believe they would maintain their positive-leaning scores of 6.
-- P5wr and FY2D: I believe there's a 50/50 chance they would increase their scores.

---

### Decision · Program_Chairs · 2026-01-26

Reject